# Preference Learning with Response Time: Robust Losses and Guarantees

**Ayush Sawarni**[*]
Stanford University
ayushsaw@stanford.edu

**Sahasrajit Sarmasarkar**[*]
Stanford University
sahasras@stanford.edu

**Vasilis Syrgkanis**
Stanford University
vsyrgk@stanford.edu

## Abstract

This paper investigates the integration of response time data into human preference learning frameworks for more effective reward model elicitation. While binary preference data has become fundamental in fine-tuning foundation models, generative AI systems, and other large-scale models, the valuable temporal information inherent in user decision-making remains largely unexploited. We propose novel methodologies to incorporate response time information alongside binary choice data, leveraging the Evidence Accumulation Drift Diffusion (EZ) model, under which response time is informative of the preference strength. We develop Neyman-orthogonal loss functions that achieve oracle convergence rates for reward model learning, matching the theoretical optimal rates that would be attained if the expected response times for each query were known a priori. Our theoretical analysis demonstrates that for linear reward functions, conventional preference learning suffers from error rates that scale exponentially with reward magnitude. In contrast, our response time-augmented approach reduces this to polynomial scaling, representing a significant improvement in sample efficiency. We extend these guarantees to non-parametric reward function spaces, establishing convergence properties for more complex, realistic reward models. Our extensive set of experiments validate our theoretical findings in the context of preference learning over images.

## 1 Introduction

Human preference feedback has emerged as a crucial resource for training and aligning machine learning models with human values and intentions. Human preference learning systems—prevalent in domains from recommender systems to robotics and natural language processing—typically solicit binary comparisons between two options and use the chosen option to infer a user's underlying utility function [BT52]. Such binary feedback is popular because it is simple, intuitive, and imposes a low cognitive load on users [BJN+22]. This paradigm underpins a wide range of applications, including tuning recommendation engines [XAY23], teaching robots personalized objectives [HIS23, WLH+22], fine-tuning large language models via reinforcement learning from human feedback (RLHF) [BJN+22, RSM+24, ZSW+19, OWJ+22b], and vision model [WDR+24, WSZ+23]. However, a single sample of binary choice conveys very limited information—it tells us which option is preferred but not how strongly it is preferred [KK19]. As a result, learning reward functions or preference models from pairwise choices can be sample-inefficient, often requiring many queries to accurately capture nuanced human preferences. This issue is exacerbated in scenarios where one outcome consistently dominates, yielding nearly deterministic choices and providing minimal insight into the degree of preference [KK19, Cli18a, Cli18b]. In practice, reward models are often learned either *implicitly*—by integrating the reward estimation directly into the policy optimization, as in

---

[*]Equal contribution.

39th Conference on Neural Information Processing Systems (NeurIPS 2025).

Direct Preference Optimization (DPO) [RSM$^+$24], which minimizes a reward-learning objective by plugging in a closed-form expression of the reward in terms of the policy—or *explicitly*—by first fitting a separate reward function to preference data and then using that function to fine-tune the policy [KWBH24, BJN$^+$22, TSS$^+$24]. Researchers have explored augmenting binary feedback with more expressive inputs like numerical ratings or confidence scores [BJN$^+$22, WBSS22], but such explicit feedback increases user effort and interface complexity [KAS21, JCB$^+$24].

One promising implicit signal is the time a human takes to make each choice. Response times are essentially free to collect and do not disrupt the user's experience [Cli18b, AFFN21]. Moreover, a rich literature in psychology and neuroscience suggests a strong link between decision response times and the strength of underlying preferences [KK19, Tye79]. In particular, faster decisions tend to indicate clearer or stronger preferences, whereas slower decisions often suggest the person found the options nearly equally preferable [KK19, AFFN21]. This inverse relationship between decision time and preference strength has been documented in cognitive experiments and is quantitatively modeled by drift-diffusion models (DDMs) of decision making [WvdMG07a, PHS05]. DDMs interpret binary choices as the result of an evidence accumulation process: when one option has a much higher subjective value, evidence accumulates quickly toward that option, leading to a fast and confident choice; conversely, if the options are nearly tied, the accumulation is slow, resulting in a longer deliberation time [WvdMG07a, BKM$^+$23, PHS05]. However, since DDMs do not admit a tractable differentiable solution for inference, researchers have proposed differentiable approximations to the response time likelihood to make them suitable for Gaussian process regression [SLBK24, KLS$^+$23].

Recent work in preference-based linear bandits has shown that integrating response times with choices—using a lightweight EZ-diffusion model from cognitive psychology—leads to substantially more data-efficient learning compared to choice-only approaches [LZR$^+$24]. These studies confirm that response times can serve as a powerful additional feedback signal, providing valuable information that boosts sample efficiency in inferring human preferences. While the prior works have broken important new ground, their scope has been somewhat limited. [LZR$^+$24] focuses on an active preference-based linear bandit in which the algorithm controls the query distribution and assumes a linear reward function. This setup does not cover more general human-in-the-loop settings—such as training large generative models—where preference data are often collected passively and the true reward function can be highly complex. In many state-of-the-art applications (e.g., RLHF for LLMs or alignment of diffusion models), feedback is gathered on model outputs drawn from a broad distribution, rather than by actively choosing each query, and the reward model is usually nonlinear, implemented as a deep neural network.

Our technical innovation centers on a Neyman-orthogonal loss function that integrates binary choice outcomes with response time observations. Neyman orthogonality ensures that small errors in estimating the response-time model do not bias the reward learning, yielding fast convergence rates. We prove that our method achieves the same asymptotic rate as an oracle that knows the true expected response time for every query, and we demonstrate in experiments on image-preference benchmarks that it significantly outperforms preference-only baselines.

**Our Contributions:**

- We propose a *Neyman-orthogonal loss* that jointly leverages response-time signals (via the EZ diffusion model) and binary preference data to estimate reward functions. This construction (i) enables integration of cognitive DDM insights into standard machine-learning-from-human-feedback frameworks, and (ii) yields significant empirical and theoretical improvements over classical MLE-based log-loss estimator that uses only preference data.

- For *linear* reward models, we derive asymptotic variance bounds showing that the estimation error of the log-loss estimator scales *exponentially* with the norm of the true parameter, whereas our orthogonal estimator's variance grows only *linearly*. Moreover, unlike the active-query method of [LZR$^+$24], which requires carefully designed query selection, our estimator achieves better variance scaling under passive, i.i.d. queries drawn from an unknown distribution.

- We extend our analysis to *nonparametric* reward classes (e.g. RKHS and neural networks), proving finite-sample convergence bounds in which errors in the response-time estimation enter only as *second-order* terms.

- We validate our framework on synthetic and semi-synthetic benchmarks. Experiments show that our orthogonal loss consistently outperforms both the log-loss and non-orthogonal alternatives in estimating linear reward functions, three-layer neural network models, and an image-based preference learning task.

**Other Related Work**   In preference-based learning, the *dueling-bandit* is a common paradigm, where an agent selects a pair of actions each round and observes a binary preference signal [YBKJ12, YJ11, YLC+22, BBFEMPH21]. Treating each action pair as a composite arm yields a closely related *logistic-bandit* formulation, which has been studied extensively for cumulative regret guarantees [FACF20, AFC21, FAJC22, ZS23, DFE18, SDBS24].

Our setting departs from these formulations along two axes: (i) *sampling*—we analyze a supervised setting with i.i.d. queries rather than adaptive selection; and (ii) *feedback*—we leverage response times in addition to noisy pairwise comparisons. While [LZR+24] also incorporates response-time feedback, their focus is best-arm identification (BAI), which is typically easier than the supervised reward-estimation problem we study. Moreover, substituting our orthogonal loss into their sequential-elimination algorithm improves the probability of best arm identification (see Appendix D.1).

Recent work by [SLBK24] uses a GP prior on rewards and develops a variational approximate Bayesian inference approach using moment matching to refine the posterior to incorporate response time information. In particular, the paper suffers from the standard issues with GP regression and does not scale to high-dimensional (curse of dimensionality in GP regression [KBCF17, BW22, GRSH22]) and is not suitable for large-scale machine learning models. Our setting is motivated by large-scale "learning from human feedback" pipelines—e.g., fine-tuning large language or vision models—where inputs live in very high dimensions and downstream tasks require a pointwise reward estimate. We completely bypass the use of variational approximations and moment-matching techniques as our method avoids modeling the full likelihood altogether.

## 2   Notations and Preliminaries

**Preference-Learning Setup**   We adopt the standard preference-learning framework. On each trial, a user is presented with two alternatives, $X^1$ and $X^2$, drawn i.i.d. from an unknown distribution $\mathcal{D}$. The user then selects one option, which we encode as a binary preference $Y \in \{+1, -1\}$, and we record the response time $T$. We further assume that both the preference $Y$ and the response time $T$ are governed by an underlying reward function $r$. The learner's objective is to learn $r$ from the observed data $\left\{ (X_i^1, X_i^2, Y_i, T_i) \right\}_{i=1}^n$.

Numerous prior works [GADGP+24, CLB+17, RSM+24] employ the *Bradley–Terry* model [BT52] to link the binary preference $Y$ to the latent reward function $r$. In this formulation, the probability of selecting alternative $X^i$ is proportional to $\exp\!\big(r(X^i)\big)$. To capture both choice accuracy and response-time variability, we adopt the EZ diffusion model, which extends this log-odds specification by modeling response-time distributions alongside choice probabilities. By design, the EZ diffusion model reduces to the Bradley–Terry model when only choice probabilities are considered. As is common in current learning from human feedback literature [OWJ+22a, ZJJ24, KWBH24], we assume homogeneity in the samples, i.e., we assume that the reward function is uniform across samples and the feature vectors encapsulate any user heterogeneity.

**EZ diffusion model**   Given a query $X^1, X^2$, the EZ diffusion model [WVDMG07b, BKM+23] treats decision making as a drift-diffusion process with drift $\mu = r(X^1) - r(X^2)$ and noise $B(\tau) \sim \mathcal{N}(0, \tau)$. After an initial encoding delay $t_{\mathrm{nd}}$, evidence accumulates as $E(\tau) = \mu\,\tau + B(\tau)$ until it first reaches one of two symmetric absorbing barriers at $+a$ or $-a$. The response time $T$ and the preference $Y$ is given by

$$T = \min\{\tau > 0 : E(\tau) \in \{\pm a\}\} \qquad Y = \begin{cases} 1, & \text{if } E(T) = a, \\ -1, & \text{if } E(T) = -a. \end{cases}$$

In most applications, one observes the total response time $t_{\mathrm{nd}} + T$, where $t_{\mathrm{nd}}$, captures the non-decision time required to perceive and encode the query. In vision-based preference tasks, $t_{\mathrm{nd}}$ is often treated as a constant [Cli18a, YK23], whereas in language or more complex cognitive tasks it may depend on the properties of the text [BSH24, VT16]. The reward difference $r(X^1) - r(X^2)$

reflects the strength of preference for a given query, while the barrier $a$ governs the conservativeness inherent to both the task and individual characteristics [VT16]. For simplicity in notation, we use $X$ to denote the pair $(X^1, X^2)$ Further, we overload $r$ and say $r(X) := r(X^1) - r(X^2)$ to denote the difference of the rewards from choices $X^1$ and $X^2$. The EZ-diffusion model implies the following key expressions as computed in [PHS05]:

$$P(Y = 1|X) = \frac{1}{1 + \exp(-2a\,r(X))} \qquad \mathbb{E}[T|X] = \begin{cases} \frac{a \tanh(a\,r(X))}{r(X)}, & \text{if } r(X) \neq 0, \\ a^2, & \text{otherwise} \end{cases} \qquad (1)$$

In applications to machine learning, both $t_{\mathrm{nd}}$ and $a$ may be informed by extensive psychology and economics literature [vRO09, Cli18a, WVDMG07b, BSH24, FNSS20, XCSWC24]. For clarity, we treat these parameters as known, fixing $a = 1$, in the main text and defer their estimation and uncertainty analysis to Appendix A. In that appendix, we show that when $a$ is unknown one may equivalently learn the scaled reward function $r(X)/a$ with identical guarantees, and that our proposed loss is only second-order sensitive to misspecification of $t_{\mathrm{nd}}$.

Note that if we ignore response times and consider only the preferences $Y$, the EZ-diffusion model reduces exactly to the *Bradley-Terry* model. Further, combining the choice and timing expressions in (1) (and setting $a = 1$) gives a convenient identity

$$r(X) = \frac{\mathbb{E}[Y \mid X]}{\mathbb{E}[T \mid X]}. \qquad (2)$$

**Additional Notations:** Now that the model is defined, we introduce notation to distinguish true functions from their estimators: we write $r_o(X)$ for the true reward-difference and $\hat{r}(X)$ for its estimate, and similarly $t_o(X) := \mathbb{E}[T \mid X]$ with estimate $\hat{t}(X)$. We further define norms $\|f(\cdot)\|_{L_2(\mathcal{D})}$ and $\|f(\cdot)\|_{L_1(\mathcal{D})}$ as $\sqrt{\mathbb{E}_{X \sim \mathcal{D}}[f(X)^2]}$ and $\mathbb{E}_{X \sim \mathcal{D}}[|f(X)|]$ respectively. We also define the random variable $Z$ as the tuple $Z := (X, Y, T)$.

**Preference-Only Learning** Estimating the reward function $r(\cdot)$ from binary preferences $Y \in \{-1, +1\}$ reduces to logistic regression, since $P(Y = 1 \mid X) = (1 + \exp(-2\,r(X)))^{-1}$. One can compute the maximum-likelihood estimate, or equivalently minimize the logistic loss:

$$\mathcal{L}^{\mathtt{logloss}}(r) = \mathbb{E}\big[\log\big(1 + \exp(-2\,Y\,r(X))\big)\big]. \qquad (3)$$

## 3 Incorporating Response Time

Using the identity in (2), a natural starting point is the "naive" squared-error loss

$$\mathcal{L}^{\mathtt{non-ortho}}(r; t) = \mathbb{E}\left[(Y - r(X)t(X))^2\right], \qquad (4)$$

where $t(X)$ estimates $t_0(X) := \mathbb{E}[T \mid X]$. However, this formulation suffers from a few serious drawbacks. Estimation of $r_o$ is highly sensitive to any error in estimating $t_o$. Moreover the function is often hard to estimate even when the reward function $r_o$ is linear. Second, even if $r_0$ is linear, the mapping $t_o(X) = \frac{\tanh(r_o(X))}{r_o(X)}$ is highly nonlinear , and the response time $T$ does not admit a simple closed-form density in terms of $r(X)$. Moreover, $T$ is also sensitive to the $t_{\mathrm{nd}}$ assumed in the model. For these reasons, the loss in (4) is generally impractical.

The function $t(\cdot)$ acts as a *nuisance* parameter in the naive loss (4) and any error in $t(\cdot)$ directly contaminates the estimate of $r_o(\cdot)$. To address this, we design a modified loss that is *Neyman-orthogonal* to $t$, eliminating its first-order effect on the gradient with respect to $r$. In the next section, we review the Neyman-orthogonality conditions and present our orthogonal loss.

### 3.1 Neyman-orthogonality and Orthogonal Statistical learning.

We consider a population loss $L(\theta, g) = \mathbb{E}\big[\ell(\theta, g; Z)\big]$, where $\theta$ is the target parameter and $g$ is a nuisance parameter. The loss is said to be *Neyman-orthogonal* at the true pair $(\theta_0, g_0)$ if its mixed

directional derivative [1] vanishes:

$$D_g \, D_\theta \, L(\theta_0, g_0)[h, k] \;=\; 0 \quad \text{for all directions } h \text{ and } k. \tag{5}$$

This condition ensures that errors $g - g_o$ have zero first-order impact on the estimator of $\theta$, so that any error from $g$ influences estimation of $\theta_o$ only at a higher order (e.g., quadratic).

## 3.2 Orthogonal Loss for Preference learning

To prevent errors in estimating the decision-time function $t$ from biasing the reward estimate $r$, we define the orthogonalized loss

$$\mathcal{L}^{\texttt{ortho}}(r; \mathfrak{r}, t) = \mathbb{E}\left[ \left( Y - (T - t(X))\mathfrak{r}(X) - r(X)t(X) \right)^2 \right], \tag{6}$$

where $\mathfrak{r}$ is a preliminary estimator of the true reward function $r_o(\cdot)$. In Lemma 3.1 we prove that $\mathcal{L}^{\texttt{ortho}}$ satisfies Neyman-orthogonality with respect to the nuisance pair $g = (\mathfrak{r}, t)$. Crucially, $\mathfrak{r}$ need only be a rough initial estimate (e.g. via logistic loss), since first-order errors in $\mathfrak{r}$ are automatically corrected in $\mathcal{L}^{\texttt{ortho}}$. As shown in Section 5, this yields a final estimator for $r$ that is robust to substantial nuisance-estimation error.

**Lemma 3.1.** *The population loss $\mathcal{L}^{\texttt{ortho}}$ is Neyman-orthogonal with respect to nuisance $g := (\mathfrak{r}, t)$ i.e. $D_g D_r \mathcal{L}^{\texttt{ortho}}(r_o; g_o)[r - r_o, g - g_o] = 0 \quad \forall r \in \mathcal{R} \quad \forall g \in \mathcal{G}$.*

*Proof.* Let $\ell(\cdot)$ be the pointwise evaluation of $\mathcal{L}^{\texttt{ortho}}$ at a data point $Z = (X, Y, T)$. A direct calculation gives

$$D_g D_r \, \ell\big(r, g_o; X, Y, T\big)[r - r_o, \; g - g_o] = 2\big(-Y + T \, r_o(X)\big)\big(t(X) - t_o(X)\big)\big(r(X) - r_o(X)\big)$$
$$+ 2\big(T \, t_o(X) - t_o(X)^2\big)\big(\mathfrak{r}(X) - r_o(X)\big)\big(r(X) - r_o(X)\big).$$

Because $r_o(X) = \dfrac{\mathbb{E}[Y \mid X]}{t_o(X)}$, we have $\mathbb{E}[-Y + T \, r_o(X) \mid X] = 0$. Similarly, by definition of $t_o$, $\mathbb{E}\big[T \, t_o(X) - t_o(X)^2 \mid X\big] = 0$. Taking expectations of the directional derivative, therefore yields

$$\mathbb{E}\big[ D_g D_r \, \ell\big(r, g_o; X, Y, T\big)[r - r_o, \; g - g_o] \big] = 0,$$

which establishes the claimed Neyman-orthogonality. $\qquad\square$

We present a Meta-Algorithm to estimate the reward model using nuisance functions $\mathfrak{r}(\cdot)$ and $t(\cdot)$.

---

**Input:** $\mathcal{S} = \{(X_i^{(1)}, X_i^{(2)}, Y_i, T_i)\}_{i=1}^n$ **Goal:** Estimate reward model $\hat{r}(\cdot)$

1: Compute nuisance functions $\hat{\mathfrak{r}}$ and $\hat{t}$ as an initial estimate of reward model and response time.
2: Now use these functions $(\hat{\mathfrak{r}}, \hat{t})$ as nuisance to minimize the orthogonalized loss function $\mathcal{L}^{\texttt{ortho}}$.

---

Meta-Algorithm 1: Estimate Reward Model via Orthogonal Loss

Different implementations of the Meta-Algorithm vary in how the nuisance functions $\mathfrak{r}$ and $t$ are estimated, following [FS23, CNR18, DKSM21]. In *data-splitting*, the data is split into two halves: nuisances are fitted on one half, and the orthogonal loss is minimized on the other. *Cross-fitting* generalizes this to $K$ folds, training nuisances out-of-fold and evaluating on each held-out fold before aggregating. *Data-reuse* fits both nuisance and target models on the full dataset. In the subsequent sections, we specify which variant is used for each theoretical guarantee and empirical experiment.

Furthermore, since the EZ diffusion model implies identities in (1), we may plug in $t(\cdot) = \frac{\tanh(\mathfrak{r}(\cdot))}{\mathfrak{r}(\cdot)}$ directly into $\mathcal{L}^{\texttt{ortho}}$, and the loss remains Neyman-orthogonal. This plug-in strategy offers further flexibility: one can first train the reward model $r$ (e.g. by minimizing the logistic loss on preference data), then uses the fitted $\mathfrak{r}$ as the nuisance in $\mathcal{L}^{\texttt{ortho}}$ to exploit response-time information $T$ for faster

---

[1]The directional derivative of $F \colon \mathcal{F} \to \mathbb{R}$ at $f$ in direction $h$ is defined by $D_f F(f)[h] = \frac{d}{dt} F(f + th)\big|_{t=0}$. For a bivariate functional $L(\theta, g)$, we write $D_\theta L$ and $D_g L$ to indicate differentiation w.r.t. each argument.

convergence. While our work focuses on reward estimation, this framework also supports DPO-style objectives [RSM$^+$24], as discussed in Appendix A. Treating the estimation of $t$ as a black box, our theoretical guarantees hold with only mild second-order corrections; see Appendix C for details.

In the next two sections, we present theoretical guarantees for Meta-Algorithm 1. In Section 4, we focus on linear reward models and state the results that show our orthogonal estimator achieves an exponential improvement in estimation error—as a function of the true reward magnitude—strictly outperforming the asymptotic rates of the preference-only estimator. In Section 5, we derive finite-sample bounds for general non-linear reward classes, including non-parametric estimators. These bounds essentially recover the oracle rates—that is, the rates one would attain if the true average response-time function $t_o(\cdot)$ were known and the naive loss $\mathcal{L}^{\texttt{non-ortho}}$ were used.

# 4 Asymptotic Rates for Linear Reward Function

We now restrict to the linear class $\mathcal{R} = \{x \mapsto \langle x, \theta \rangle : \theta \in \mathbb{R}^d\}$, so that $r_\theta(X) = \langle \theta, X \rangle$ and the true reward, $r_o(X) = \langle \theta_o, X \rangle$. Our goal is to estimate $\theta_o$.

**Preference-only estimator.** Let $\hat{\theta}_{\texttt{log}}$ minimize the empirical version of the logistic loss in (3). Under the condition that $\mathbb{E}\left[\sigma(-2\langle \theta_0, X \rangle)\,\sigma(2\langle \theta_0, X \rangle)\,XX^\top\right]$ is invertible, standard argument such as the ones in [FK85] (see Appendix B for derivation) yield

$$\sqrt{n}\left(\hat{\theta}_{\texttt{log}} - \theta_0\right) \xrightarrow{d} \mathcal{N}\Big(0,\ \left[4\,\mathbb{E}\left[\sigma(-2\langle \theta_0, X \rangle)\,\sigma(2\langle \theta_0, X \rangle)\,XX^\top\right]\right]^{-1}\Big). \tag{7}$$

**Orthogonal estimator.** Let $\hat{\theta}_{\texttt{ortho}}$ denote the resulting estimator from Meta-Algorithm 1 (either with cross-fitting or data-splitting).

**Theorem 4.1.** *Let $\hat{\theta}_{\texttt{ortho}}$ minimize the orthogonal loss $\mathcal{L}^{\texttt{ortho}}$. If $\mathbb{E}\left[t_0(X)^2 XX^\top\right]$ is invertible, then*

$$\sqrt{n}\left(\hat{\theta}_{\texttt{ortho}} - \theta_o\right) \xrightarrow{d} \mathcal{N}\left(0, \Sigma\right)$$

*where $\Sigma = \mathbb{E}\left[\left(\frac{\tanh(\langle \theta_o, X \rangle)}{\langle \theta_o, X \rangle}\right)^2 XX^\top\right]^{-2} \mathbb{E}\left[\left(\frac{\tanh(\langle \theta_o, X \rangle)}{\langle \theta_o, X \rangle}\right)^3 XX^\top\right].$*

Theorem 4.1 is proven in Appendix B using techniques developed in [CNR18] for asymptotic statistics for debiased estimators. Furthermore, the asymptotic variance of $\hat{\theta}_{\texttt{ortho}}$ is point-wise smaller than that of $\hat{\theta}_{\texttt{log}}$, and this continues to hold for any barrier height $a$. This follows from the fact $4\,\sigma(2x)\sigma(-2x) \leq \left(\frac{\tanh(x)}{x}\right)^2$ for all $x \geq 0$ (see Appendix B). Because $\frac{\tanh(x)}{x}$ decays polynomially with $|x|$ while $\sigma(x)\sigma(-x)$ decays exponentially, the variance of the orthogonal estimator is exponentially smaller than the log-loss estimator.

Our asymptotic guarantee differs fundamentally from [LZR$^+$24, Theorem 3.1]. Li et al. establish point-wise convergence for a fixed query $x$ as the number of observations of that specific query approaches infinity, whereas our result is framed in terms of the overall sample size $n$ accumulated by the learner. Basing the limit on $n$ rather than on per-query counts better reflects how data are gathered in practical preference-learning scenarios.

Moreover, while this guarantee also covers the setting described in [LZR$^+$24], the guarantees are significantly tighter. In particular the variance of their estimator decays with $\left(\min_{x \in \mathcal{X}_{\text{sample}}} \frac{\tanh(\langle \theta_o, x \rangle)}{\langle \theta_o, x \rangle}\right)^{-1}$ where $\mathcal{X}_{\text{sample}}$ denotes the set of distinct pairs of queries $x$. In contrast, in our case, this term is inside expectation and thus results in stronger guarantees.

One may naturally ask how our rates depend on the accuracy of the nuisance estimates $\hat{\mathfrak{r}}(\cdot)$ and $\hat{t}(\cdot)$. We prove that the asymptotic guarantees hold, provided

$$\left\|\hat{t} - t_o\right\|_{L_2(\mathcal{D})} = o\left(n^{-\beta_1}\right), \qquad \beta_1 \geq \tfrac{1}{4}, \qquad \left\|\hat{\mathfrak{r}} - r_o\right\|_{L_2(\mathcal{D})} = o\left(n^{-\beta_2}\right), \qquad \beta_2 \geq \tfrac{1}{2} - \beta_1. \tag{8}$$

For linear rewards, one can achieve the required "slow" convergence rates for estimating $t$ by applying kernel ridge regression with a Sobolev kernel associated with the RKHS $W^{s,2}$ [AF03, Wai19].

Alternatively, one may first fit $\hat{\mathfrak{r}}$ via logistic-loss minimization and then set $\hat{t}(X) = \frac{\tanh(\hat{\mathfrak{r}}(X))}{\hat{\mathfrak{r}}(X)}$. Linearity of $r$ and the Lipschitz continuity of the mapping $x \mapsto \frac{\tanh(x)}{x}$ then ensure the required slow-rate bounds (see Appendix C).

## 5  Finite Sample Guarantees for General Reward Functions

We denote the joint nuisance pair as $g = (\mathfrak{r}, t)$. In this section we bound the estimation error $\|\hat{r} - r_o\|_{L_2(\mathcal{D})}$ in terms of the population pseudo-excess risk defined as $\mathcal{L}^{\text{ortho}}(\hat{r}; \hat{g}) - \mathcal{L}^{\text{ortho}}(r_o, \hat{g})$ plus higher-order terms depending on nuisance estimation error. We assume $r_o \in \mathcal{R}$ and $t_o \in \mathcal{T}$, and let $\mathcal{G} = \mathcal{R} \times \mathcal{T}$ denote the joint nuisance class (and $g \in \mathcal{G}$). Further let $\text{star}\{\mathcal{X}, x'\} := \{(1-t)x + tx' : x \in \mathcal{X}; t \in [0,1]\}$. Further, we adopt the standard $L_p$ norm for functions i.e.

$$\|f\|_{L_2(\mathcal{D})} = \sqrt{\mathbb{E}_{X \sim \mathcal{D}}[f(X)^2]}, \qquad \|f\|_{L_1(\mathcal{D})} = \mathbb{E}_{X \sim \mathcal{D}}\big[|f(X)|\big], \qquad \|f\|_{L_\infty} = \sup_x |f(x)|.$$

Next we define a pre-norm $(\mathcal{R}, \alpha)$ (need not satisfy triangle inequality) on the nuisance function $g(\cdot)$ to first present a general guarantee in term of this norm for any function class, and then further bound it with $L_2(\mathcal{D})$ norm, attaining different rates for different function classes such as RKHS balls and finite-VC subclasses following the general theorem and in Appendix C.

$$\|g\|^2_{(\mathcal{R}, \alpha)} = \sup_{r \in \mathcal{R}} \left( \frac{\|\mathfrak{r}(\cdot) t(\cdot) r(\cdot)\|_{L_1(\mathcal{D})}}{\|r\|^{1-\alpha}_{L_2(\mathcal{D})}} + \frac{\|t(\cdot)^2 r(\cdot)\|_{L_1(\mathcal{D})}}{\|r\|^{1-\alpha}_{L_2(\mathcal{D})}} \right) \tag{9}$$

While Neyman orthogonality ensures higher-order dependence on nuisance error, defining the pre-norm this way gives additional robustness with respect to errors in $\mathfrak{r}$. The product $\mathfrak{r}(\cdot) t(\cdot)$ couples the estimation errors of $\hat{\mathfrak{r}}$ and $\hat{t}$: even if $\mathfrak{r}$ is poorly estimated—say, under large reward magnitudes—a sufficiently accurate estimate of $t$ can keep the product of errors small, yielding low nuisance error in the $(\mathcal{R}, \alpha)$ norm.

**Theorem 5.1.** *Suppose $\hat{r}$ minimizes the orthogonal population loss $\mathcal{L}^{\text{ortho}}$ and satisfies*

$$\mathcal{L}^{\text{ortho}}(\hat{r}; \hat{g}) - \mathcal{L}^{\text{ortho}}(r_o; \hat{g}) \leq \epsilon(\hat{r}, \hat{g}).$$

*Let $S$ be an absolute bound on $r_o$. Then the two-stage Meta-Algorithm 1 guarantees*

$$\|\hat{r} - r_o\|^2_{L_2(\mathcal{D})} \leq 4S^2 \epsilon(\hat{r}, \hat{g}) + 4 S^{\frac{4}{1+\alpha}} \|\hat{g} - g_o\|^{\frac{4}{1+\alpha}}_{(\mathcal{R}, \alpha)}.$$

The proof of the theorem follows by instantiating [FS23, Theorem 1]. The details can be found in Appendix C. Now we instantiate the above theorem for specific function classes.

### 5.1  Nuisance estimation error $\|\hat{g} - g_o\|_{(\mathcal{R}, \alpha)}$

Let $\|r\|_{\mathcal{R}}$ denote the RKHS norm of $r$. First, if $r \in \mathcal{R}$ lies in an RKHS ball of bounded radius and with a kernel whose eigenvalues decay as $j^{-1/\alpha}$, then by [MN10] we have $\|r\|_\infty \leq C \|r\|^\alpha_{\mathcal{H}} \|r\|^{1-\alpha}_{L_2(\mathcal{D})}$, which in turn implies $\|g\|_{(\mathcal{R}, \alpha)} \leq C \sqrt{\|t(\cdot)\mathfrak{r}(\cdot)\|_{L_1(\mathcal{D})} + \|t(\cdot)^2\|_{L_1(\mathcal{D})}}$.

Second, if we have $\frac{\|t\|_{L_4(\mathcal{D})}}{\|t\|_{L_2(\mathcal{D})}} \leq C_{2 \to 4}$ and $\frac{\|\mathfrak{r}\|_{L_4(\mathcal{D})}}{\|\mathfrak{r}\|_{L_2(\mathcal{D})}} \leq C_{2 \to 4}$, then setting $\alpha = 0$ and applying Cauchy–Schwarz gives

$$\|g\|^2_{(\mathcal{R}, \alpha)} \leq C^2_{2 \to 4} \left( \|t\|_{L_2(\mathcal{D})} \cdot \|\mathfrak{r}\|_{L_2(\mathcal{D})} + \|t\|^2_{L_2(\mathcal{D})} \right).$$

Under these two cases, any bound $\zeta$ on $\|\hat{t} - t_o\|_{L_2(\mathcal{D})}$ and $\|\hat{\mathfrak{r}} - r_o\|$ shows up as $\zeta^{\frac{4}{1+\alpha}}$ in reward estimation error.

*Remark* 5.2. The finite-sample excess-risk bounds in Corollaries 5.3 and 5.4 are obtained via a critical-radius argument and Talagrand's concentration inequality, both of which require the loss to be uniformly bounded point-wise. However, our original (pointwise) orthogonal loss: $\ell^{\text{ortho}}(r, g; Z) = \big(Y - (T - t(X))\mathfrak{r}(X) - r(X)t(X)\big)^2$, can be unbounded when the decision time $T$ is unbounded, violating these assumptions.

To remedy this, in Appendix C, we introduce a simple modification where we cap the contribution of any single decision time $T$ at a large constant $\breve{B}$. This puts an additional bias on the $\ell_2$ error in Theorem 5.1 decaying fast with threshold $\breve{B}$. Moreover, in any real-world use case, all the recorded response times would always be bounded; this modeling choice aligns with a real-world application of the framework. The effect of capped-loss construction on finite-sample rates appear in Appendix C.

## 5.2 Bounding excess-risk $\epsilon(\hat{r}, \hat{g})$

We consider two nuisance estimation schemes for Meta-Algorithm 1: *data-splitting* (independent nuisance and target estimation) and *data-reuse* (joint estimation). We start by defining critical radius of a function class and provide the guarantees in terms of the critical radius. Further, let $\ell^{\text{ortho}}$ denote the point-wise loss version of population loss $\mathcal{L}^{\text{ortho}}$.

**Critical radius.** Following [Wai19], define the *localized Rademacher complexity* of a function class $\mathcal{F}$ by

$$\text{Rad}_n(\mathcal{F}, \delta) := \mathbb{E}_{\epsilon, z_{1:n}} \left[ \sup_{\substack{f \in \mathcal{F} \\ \|f\|_{L_2(\mathcal{D})} \leq \delta}} \left| \frac{1}{n} \sum_{i=1}^{n} \epsilon_i \, f(z_i) \right| \right].$$

The *critical radius* $\delta_n$ is the smallest $\delta > 0$ satisfying $\text{Rad}_n(\mathcal{F}, \delta) \leq \delta^2$.

We now present the rates in terms of moment function $M(\zeta)$ which is an upper bound on moment of $\mathbb{E}[T^\zeta \mid X]$ assuming that $r(X) \leq S$ for every $\zeta > 1$. From [WY08], we know that every $\zeta^{\text{th}}$ moment exists. One can choose $\zeta$ to get the tightest bound.

**Corollary 5.3** (Data-splitting)**.** *Let $\delta_n$ be the critical radius of the star-shaped class*

$$\text{star}\{ r - r_o \colon r \in \mathcal{R}, 0 \},$$

*and define $\delta_n^{\text{ds}} = \max\{\delta_n, \sqrt{c/n}\}$ for some constant $c > 0$. Then under data-splitting, with probability at least $1 - c_1 \exp(-c_2 n (\delta_n^{\text{ds}})^2)$, Meta-Algorithm 1 satisfies*

$$\epsilon(\hat{r}, \hat{g}) \leq 9\, S\, \breve{B} \delta_n^{\text{ds}} \, \|\hat{r} - r_o\|_{L_2(\mathcal{D})} + 10\, S^2 \breve{B}\, (\delta_n^{\text{ds}})^2,$$

*for universal constants $c_1, c_2 > 0$. Consequently for every $\zeta \geq 1$,*

$$\|\hat{r} - r_o\|_{L_2(\mathcal{D})}^2 \leq \text{poly}(S) \left( \breve{B} \left(\delta_n^{\text{ds}}\right)^2 + \left( \frac{M(\zeta)}{\zeta \breve{B}^{\zeta - 1}} \right)^2 \right) + 4\, S^{\frac{4}{1+\alpha}} \, \|\hat{g} - g_o\|_{(\mathcal{R}, \alpha)}^{\frac{4}{1+\alpha}},$$

*holds with the same probability.*

For the case of data-reuse we have,

**Corollary 5.4** (Data-reuse)**.** *Let $\delta_n$ be the critical radius of*

$$\text{star}\{ \ell^{\text{ortho}}(r, g; \cdot) - \ell^{\text{ortho}}(r_o, g; \cdot) \colon r \in \mathcal{R}, \ g \in \mathcal{G} \},$$

*and define $\delta_n^{\text{dr}} = \max\{\delta_n, \sqrt{c/n}\}$ for some constant $c > 0$. Then under data-reuse, with probability at least $1 - c_1 \exp(-c_2 n (\delta_n^{\text{dr}})^2)$, Meta-Algorithm 1 satisfies*

$$\epsilon(\hat{r}, \hat{g}) \leq 9\, S\, \delta_n^{\text{dr}} \, \|\hat{r} - r_o\|_{L_2(\mathcal{D})} + 10\, (\delta_n^{\text{dr}})^2,$$

*for universal constants $c_1, c_2 > 0$. Consequently for every $\zeta \geq 1$,*

$$\|\hat{r} - r_o\|_{L_2(\mathcal{D})}^2 \leq \text{poly}(S) \left( (\delta_n^{\text{dr}})^2 + \left( \frac{M(\zeta)}{\zeta \breve{B}^{\zeta - 1}} \right)^2 \right) + 4\, S^{\frac{4}{1+\alpha}} \, \|\hat{g} - g_o\|_{(\mathcal{R}, \alpha)}^{\frac{4}{1+\alpha}},$$

*holds with the same probability.*

The full proof of the critical-radius bounds and further discussion appear in Appendix C[2]. Our analysis follows [Wai19, Theorem 14.20], leveraging uniform law for Lipschitz loss functions.

The critical radius differs between data-splitting and data-reuse: with data reuse, each observation $Z = (X, Y, T)$ influences both the reward model $r(\cdot)$ and the nuisance $g(\cdot)$, creating conditional dependence that increases the critical radius and slows convergence, whereas under data splitting the two estimates remain independent, yielding a smaller radius and faster rates.

---

[2]Above bounds in Corollary 5.4 and Corollary 5.3 are loose in $S$ and can be tightened via a careful analysis.

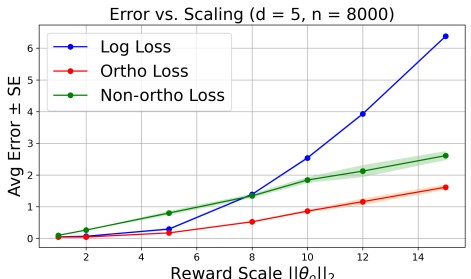
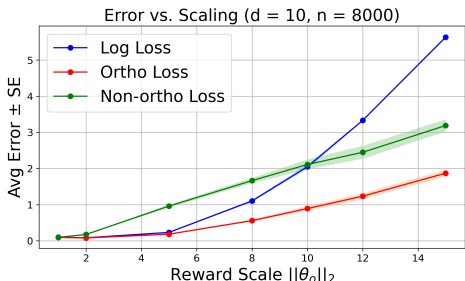

Figure 1: Performance of the linear-reward model as the true parameter magnitude $\|\theta_0\|$ varies. Left: $d = 5$; right: $d = 10$.

## 6 Experiments

We evaluate three settings: (a) linear reward models, (b) neural network–parameterized rewards, and (c) a semi-synthetic text-to-image preference dataset. In addition, Appendix D.1 applies our orthogonal loss within the sequential-elimination algorithm of [LZR+24] for best-arm identification, where it outperforms their algorithms.[3]

**Linear rewards.** We evaluate on synthetic data where each query pair $(X^1, X^2)$ is drawn uniformly from the unit-radius sphere and the true reward is $r_0(X) = \langle \theta_o, X \rangle$ with $\|\theta_o\|_2 = B$. Preferences $Y$ and response times $T$ are generated via the EZ diffusion model. For each $B$, we draw $\theta_o$ at random, generate 10 independent datasets, and fit $\theta$ by minimizing the logistic loss, non-orthogonal loss and orthogonal loss. Further to estimate the nuisance parameter $\mathfrak{r}(\cdot)$ and $t(\cdot)$ for orthogonal and non-orthogonal loss, we use logistic regression and a 3 layered neural network respectively. We report the average error $\|\hat\theta - \theta_o\|_2$ as we vary $B$ in Figure 1. Full experimental details are provided in Appendix D. We observe that the estimation error under the logistic loss grows rapidly with $B$ and the orthogonal loss $\mathcal{L}^{\text{ortho}}$ consistently outperforms the other two losses.

**Non-linear rewards—neural networks** We generate synthetic data from random three-layer neural networks with sigmoid activations, fixed input dimension, and hidden-layer widths. For each training size $N$, we sample a new network (details in Appendix D) as the true reward model and draw $N$ query pairs $X_1, X_2$ uniformly from the unit sphere. We evaluate all three losses—logistic, non-orthogonal, and orthogonal—and for the orthogonal loss we compare both a simple data-split implementation and a data-reuse implementation. Figure 2 reports the mean squared error (and $\pm$ standard error) of the estimated reward under each of the three loss functions and the corresponding policy regret after thresholding $\hat r$ into a binary decision. Regret measures the gap between the learned policy and the optimal binary policy. Denoting by $\hat X_i$ the option selected by our policy for query $i$, the regret over $M$ new queries is

$$\text{Regret} = \sum_{i=1}^{M} \Big( \max\{r_o(X_i^1),\, r_o(X_i^2)\} \, - \, r_o(\hat X_i) \Big), \qquad \hat X_i = \begin{cases} X_i^1, & \hat r(X_i^1) \geq \hat r(X_i^2), \\ X_i^2, & \text{otherwise.} \end{cases}$$

**Text-to-image preference learning.** We evaluate our approach on a real-world text-to-image preference dataset - Pick-a-pick [KPS+23], which contains an approx 500k text-to-image dataset generated from several diffusion models. Furthermore, we use the PickScore model [KPS+23] as an oracle reward function, we simulate binary preferences $Y \in \{+1, -1\}$ and response times $T$ via the EZ-diffusion process conditioned on the PickScore difference between each image-test pair. To learn the reward model we extract 1024-dimensional embeddings from both the text prompt and the generated image using the CLIP model [RKH+21]. On top of these embeddings, we train a 4-layered feed-forward neural network with hidden layers of sizes $1024, 512, 256$, under three training objectives: our proposed orthogonal loss, a non-orthogonal response-time loss, and the standard log-loss on binary preferences. Further experiment details are available in Appendix D. As

---

[3]The experiment code is available in `https://github.com/sawarniayush/Preference-Learning-with-Response-Time`.

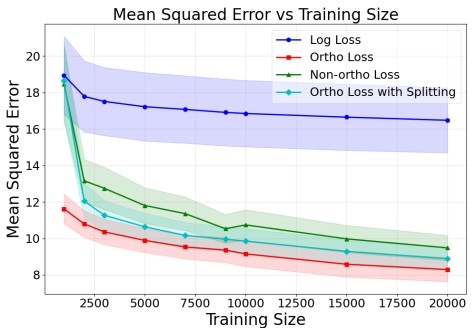 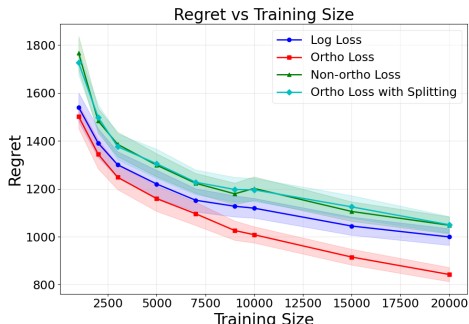

Figure 2: Left: mean-squared error ($\pm$ standard error); right: cumulative regret ($\pm$ standard error) over $M = 3000$ new queries on randomly sampled non-linear (neural network) reward models, both plotted against training-set size $N$.

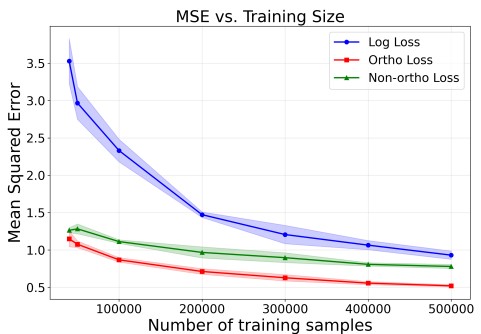 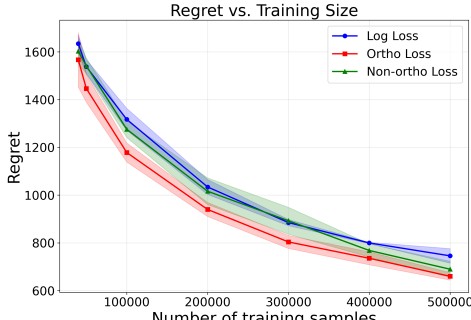

Figure 3: Left: mean-squared error ($\pm$ standard deviation); right: cumulative regret ($\pm$ standard deviation) over $M = 10000$ new queries on the Pick-a-Pic text-to-image task, both plotted against training-set size $N$.

shown in Figure 3, the orthogonal loss consistently achieves significantly lower Mean squared error and regret compared to the non-orthogonal and log-loss baselines.

## 7 Conclusion and future work

Our work proposes a Neyman-orthogonal loss function that jointly learns over the preferences and response time to estimate reward functions. We show that it better estimates the reward model theoretically and empirically in semi-synthetic setups. Possible future directions might include:

**Extension to bandit setup** : Dueling bandits model an online setting where the learner queries arm pairs and observes a noisy binary preference. Extending this framework to incorporate response time as an auxiliary feedback signal—and adapting our supervised loss to such adaptive querying for regret minimisation —presents an interesting direction for future work.

**Experiments with true response time data** : Our experiments assume a homogeneous reward model and synthetically generated response times. Evaluating on real-world response time data, which may be noisy and population-dependent (with varying barriers $a$ and true rewards $r_o(\cdot)$), could provide valuable insights into model robustness.

**DPO-style loss [RSM$^+$24]** : Beyond reward estimation, our framework can be extended to direct policy learning via a DPO-style objective by replacing the reward model with a policy parameterization. Adapting the orthogonal loss $\mathcal{L}^{\text{ortho}}$ for this purpose is a promising avenue (see Appendix A.6).

## Acknowledgments

Vasilis Syrgkanis is supported by NSF Award IIS-2337916. Ayush Sawarni is partially supported by NSF Award IIS-2337916.

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

# A Discussion on Barrier $a$ and Non-Decision Time $t_{\mathrm{nd}}$

## A.1 Properties of the EZ Diffusion Model

Recall the EZ diffusion model for a decision between two options $X_1, X_2$, with drift $\mu = r(X_1) - r(X_2)$ and symmetric absorbing barriers at $\pm a$. Writing $X = (X_1, X_2)$ and overloading $r(X) = r(X_1) - r(X_2)$, the choice and response-time moments are given by [WvdMG07a]:

$$P\big(Y = 1 \mid X\big) = \frac{1}{1 + e^{-2a\,r(X)}}, \quad E\big[Y \mid X\big] = \tanh\big(a\,r(X)\big), \quad \mathrm{Var}\big(Y \mid X\big) = 1 - \tanh^2\big(a\,r(X)\big),$$

$$E\big[T \mid X\big] = \begin{cases} \dfrac{a\,\tanh\big(a\,r(X)\big)}{r(X)}, & r(X) \neq 0, \\ a^2, & r(X) = 0, \end{cases} \quad \mathrm{Var}\big(T \mid X\big) = \begin{cases} \dfrac{a\big(e^{4a\,r(X)} - 1 - 4a\,r(X)e^{2a\,r(X)}\big)}{r(X)^3\big(e^{2a\,r(X)} + 1\big)^2}, & r(X) \neq 0, \\ \dfrac{2a^4}{3}, & r(X) = 0. \end{cases}$$

## A.2 Loss Functions for General Barrier $a$

When $a$ is known, a natural *non-orthogonal* baseline loss is

$$\mathcal{L}^{\mathtt{non-ortho}}(r;\, t) = \mathbb{E}\Big[\big(Y - \tfrac{r(X)\,t(X)}{a}\big)^2\Big],$$

where $t(X) \approx E[T \mid X]$. This also results in the following *orthogonal* loss

$$\mathcal{L}^{\mathtt{ortho}}(r;\, \mathfrak{r},\, t) = \mathbb{E}\Big[\big(Y - (T - t(X))\,\tfrac{\mathfrak{r}(X)}{a} - \tfrac{r(X)\,t(X)}{a}\big)^2\Big], \tag{10}$$

where $\mathfrak{r}(\cdot)$ is a crude estimate of the reward function $r_o(\cdot)$ (which can be calculated by minimizing the loss using the log loss) specified below

$$\mathcal{L}^{\mathtt{logloss}}(r) = \mathbb{E}\big[\log\big(1 + \exp(-2\,Y\,a\,r(X))\big)\big]. \tag{11}$$

## A.3 Reward Learning When $a$ Is Unknown

If the barrier $a$ is unknown, one can still estimate the scaled reward $r(\cdot)/a$ with identical guarantees. Note that this suffices for any standard machine learning tasks involving learning from human feedback. Indeed, minimizing the log-loss (11) yields an estimate of $a\,r_o(\cdot)$, and hence of $r_o(\cdot)$ up to scale. However, the loss (10) cannot be applied directly because it requires the nuisance $\mathfrak{r}(\cdot)/a$, which is not identifiable without knowing $a$.

Instead, we introduce a new nuisance pair $(y, t)$, where $y(X)$ approximates $E[Y \mid X]$. We also define the notation $y_o(X) := \mathbb{E}[Y \mid X]$. One can then define an alternative orthogonal loss,

$$\mathcal{L}^{\mathtt{ortho-2}}(r;\, y,\, t) = \mathbb{E}\Big[\big(Y - (T - t(X))\tfrac{y(X)}{t(X)} - \tfrac{r(X)\,t(X)}{a}\big)^2\Big]. \tag{12}$$

Equation (12) is also Neyman-orthogonal with respect of $(y, t)$ as we show in the lemma below 3.1. Moreover, the error rate guarantees for 10 and 12 are identical for the linear case, as shown in Appendix B.

**Lemma A.1.** *The population loss $\mathcal{L}^{\mathtt{ortho-2}}$ is Neyman-orthogonal with respect to nuisance $g := (y, t)$ i.e. $D_g D_r \mathcal{L}^{\mathtt{ortho-2}}(y_o; g_o)[r - r_o, g - g_o] = 0 \quad \forall r \in \mathcal{R} \quad \forall g \in \mathcal{G}$.*

*Proof.* Let $\ell(\cdot)$ be the pointwise evaluation of $\mathcal{L}^{\mathtt{ortho-2}}$ at a data point $Z = (X, Y, T)$. A direct calculation gives

$$D_g D_r\,\ell\big(r, g_o; X, Y, T\big)[r - r_o,\, g - g_o] = \frac{-2}{a}\big(Y + y_o(X) - 2\frac{r_o(X)}{a}t_o(X)\big)\big(t(X) - t_o(X)\big)\big(r(X) - r_o(X)\big)$$

$$+ \frac{2}{a}\big(T - t_o(X)\big)\big(y(X) - y_o(X)\big)\big(r(X) - r_o(X)\big).$$

Because $\frac{r_o(X)}{a} = \frac{\mathbb{E}[Y \mid X]}{t_o(X)}$, we have $\mathbb{E}\left[Y + y_o(X) - 2\frac{r_o(X)}{a}t_o(X) \mid X\right] = 0$. Similarly, by definition of $t_o$, $\mathbb{E}[T - t_o(X) \mid X] = 0$. Taking expectations of the directional derivative, therefore yields

$$\mathbb{E}\big[D_g D_r \,\ell\big(r, g_o; X, Y, T\big)[r - r_o, \; g - g_o]\big] = 0,$$

which establishes the claimed Neyman-orthogonality. $\qquad\square$

## A.4 Experiment for varying $a$

We conduct experiments for varying barrier levels $a$ for the case where the barrier $a$ is unknown. Similar the synthetic neural network experiment in the main paper, we generate synthetic data from random three-layer neural networks with sigmoid activations, fixed input dimension ($= 10$), and hidden-layer widths $(32, 16)$. We sample a random neural network (details in Appendix D) as the true reward model and draw 2000 query pairs $X_1$, $X_2$ sampled from a spherical guassian distribution. We evaluate all three losses—logistic, non-orthogonal, and orthogonal $\mathcal{L}^{\texttt{ortho}-2}$ for different values of barrier $a$. We find that the mean squared error of the estimated reward under each of the three loss functions and the corresponding policy regret (after thresholding $\hat{r}$ into a binary decision) is consistently better for $\mathcal{L}^{\texttt{ortho}-2}$. [4]

Table 1: Mean squared error for different losses across barrier values

| Barrier $a$ | Log-loss $\mathcal{L}^{\texttt{logloss}}$ | Non-orthogonal $\mathcal{L}^{\texttt{non}-\texttt{ortho}}$ | Orthogonal $\mathcal{L}^{\texttt{ortho}-2}$ |
|---|---|---|---|
| 0.5 | 13.1047 | 10.3015 | 9.2564 |
| 0.7 | 14.8284 | 10.8872 | 9.3922 |
| 0.9 | 16.1897 | 11.2413 | 9.2729 |
| 1.1 | 17.0966 | 11.7575 | 9.5093 |
| 1.3 | 17.8099 | 11.8773 | 9.6774 |
| 1.5 | 18.4114 | 12.3013 | 9.7517 |
| 1.7 | 18.9334 | 12.4945 | 9.8271 |
| 1.9 | 19.2903 | 12.6231 | 9.8571 |
| 2.1 | 19.6200 | 12.8721 | 9.8152 |
| 2.3 | 19.8787 | 13.1132 | 9.9202 |
| 2.5 | 20.1948 | 13.1526 | 10.1428 |

## A.5 Second order dependence in errors in $t_{\mathrm{nd}}$

Theorem 5.1 directly implies the following corollary when $t_{\mathrm{nd}}$ is misspecified

**Corollary A.2.** *Let $\widetilde{t}_{\mathrm{nd}}$ be any estimate of the non-decision time satisfying*

$$\left|\widetilde{t}_{\mathrm{nd}} - t_{\mathrm{nd}}\right| \leq \epsilon.$$

*and nuisance $\hat{t}^{(\epsilon)}$ is estimated using mis-specified decision times by*

$$\widetilde{T} = T_{\mathrm{total}} - \widetilde{t}_{\mathrm{nd}},$$

*and let $\hat{r}^{(\epsilon)}$ be the reward estimator obtained by replacing $T$ with $\widetilde{T}$ in the orthogonal loss. Then under the same conditions as Theorem 5.1,*

$$\left\|\hat{r}^{(\epsilon)} - r_o\right\|^2_{L_2(\mathcal{D})} = O_\epsilon\big(\epsilon^{\frac{4}{1+\alpha}}\big) + o_n(1),$$

*where $o_n(1)$ is the estimation-error from Theorem Theorem 5.1 and $\alpha$ is as in that theorem.*

---

[4]Recall that the regret over $M$ new queries is

$$\text{Regret} = \sum_{i=1}^{M}\Big(\max\{r_o(X_i^1), \, r_o(X_i^2)\} - r_o(\hat{X}_i)\Big), \qquad \hat{X}_i = \begin{cases} X_i^1, & \hat{r}(X_i^1) \geq \hat{r}(X_i^2), \\ X_i^2, & \text{otherwise.} \end{cases}$$

Table 2: Regret for $M = 2000$ queries for different losses across different barrier values

| **Barrier** $a$ | Log-loss $\mathcal{L}^{\texttt{logloss}}$ | Non-orthogonal $\mathcal{L}^{\texttt{non-ortho}}$ | Orthogonal $\mathcal{L}^{\texttt{ortho-2}}$ |
|---|---|---|---|
| **0.5** | 0.2980 | 0.2537 | 0.2538 |
| **0.7** | 0.3437 | 0.2987 | 0.2971 |
| **0.9** | 0.3635 | 0.3000 | 0.3016 |
| **1.1** | 0.3732 | 0.3152 | 0.3044 |
| **1.3** | 0.3686 | 0.3142 | 0.3120 |
| **1.5** | 0.3807 | 0.3148 | 0.3115 |
| **1.7** | 0.3787 | 0.3155 | 0.3105 |
| **1.9** | 0.3751 | 0.3216 | 0.3037 |
| **2.1** | 0.3750 | 0.3198 | 0.3082 |
| **2.3** | 0.3772 | 0.3203 | 0.3028 |
| **2.5** | 0.3763 | 0.3171 | 0.3085 |

*Proof.* Apply Theorem 5.1 with the nuisance pair $\hat{g} = (\hat{\mathfrak{r}}, \hat{t}^{(\epsilon)})$. Observe that:

1. The misspecification $|\widetilde{t}_{\mathrm{nd}} - t_{\mathrm{nd}}| \leq \epsilon$ induces at most an $O(\epsilon)$ error in the estimated nuisance function $\widetilde{t}(X)$.

2. By Neyman-orthogonality, the orthogonal loss incurs only higher-order dependence on any nuisance error; in particular, a perturbation of order $\epsilon$ in $\hat{t}$ contributes $O(\epsilon^4)$ to the final reward-estimation error.

3. The debiasing term $(T - t(X))$ is unaffected by the shift in non-decision time, since $\widetilde{t}_{\mathrm{nd}}$ cancels in the difference.

Moreover, we can write

$$\|\hat{g} - g_o\|^2_{(\mathcal{R},\alpha)} = o_n(1) + C \sup_{r \in \mathcal{R}} \left( \frac{\left\|\epsilon^2 r(\cdot)\right\|_{L_1(\mathcal{D})}}{\|r\|^{1-\alpha}_{L_2(\mathcal{D})}} \right) \tag{13}$$

where $C$ depends only on the uniform bound $S = \|r\|_\infty$. Combining these observations with the $o_n(1)$ estimation rate from Theorem 5.1 yields the stated bound. $\qquad\square$

### A.6 Estimating the nuisance $t$ via a plug-in of $\mathfrak{r}$

A convenient way to estimate $t(\cdot)$ is to exploit the EZ-diffusion identity

$$t(X) = \frac{\tanh(\mathfrak{r}(X))}{\mathfrak{r}(X)}.$$

Substituting this directly into the orthogonal loss $\mathcal{L}^{\mathrm{ortho}}$ preserves Neyman-orthogonality. Concretely, one proceeds in two stages:

1. Fit a preliminary reward model $\hat{\mathfrak{r}}$ (for example by minimizing the logistic loss on preference data).

2. Define the plug-in nuisance

$$\hat{t}(X) = \frac{\tanh(\hat{\mathfrak{r}}(X))}{\hat{\mathfrak{r}}(X)},$$

and minimize the orthogonalized squared loss with $\hat{\mathfrak{r}}$ as the nuisance

$$\mathcal{L}^{\mathrm{ortho}}(r; \mathfrak{r}) = \mathbb{E}\left[ \left( Y - T\mathfrak{r}(X) + \tanh(\mathfrak{r}(X)) - r(X)\frac{\tanh(\mathfrak{r}(X))}{\mathfrak{r}(X)} \right)^2 \right]. \tag{14}$$

This plug-in strategy retains the first-order robustness to errors in $\mathfrak{r}$ and yields faster convergence by leveraging response-time information $T$ in the second stage.

Hence, one can extend this idea to directly learn a policy on preference data. In Direct Preference Optimization (DPO) [RSM$^+$24], one obtains the reward function in closed form from a learned policy $\pi$:

$$\mathfrak{r}(X) = \mathfrak{r}(X^1) - \mathfrak{r}(X^2) = c \left( \log \frac{\pi(X^1)}{\pi_{\text{ref}}(X^1)} - \log \frac{\pi(X^2)}{\pi_{\text{ref}}(X^2)} \right)$$

where $\pi_{\text{ref}}$ is the reference policy and $c$ is a constant. We can embed these into the two-stage procedure: 1) Train $\hat{\pi}$ by minimizing the logistic-loss (DPO-loss) on the observed preferences, yielding $\mathfrak{r}(X) = \log \frac{\hat{\pi}(X)}{\pi_{\text{ref}}(X)}$. 2) Plug $\mathfrak{r}$ into (14). Since $r$ itself is a known function of $\pi$, this directly learns a new policy that incorporates response time. Adapting $\mathcal{L}^{\text{ortho}}$ for directly learning a policy from preference data can be an interesting direction for future work.

# B Asymptotic guarantee for the linear reward model classes

## B.1 Guarantees for $\mathcal{L}^{\mathtt{logloss}}$

We now give asymptotic guarantees on the convergence of the choice-only estimator using the idea of influence functions [Hub11, Ham74]. For the choice-only estimator, the logistic regression loss function $\ell(\theta, X, Y) := \log(\sigma(2aY\langle\theta, X\rangle))$ as described in (11). [5] The informal lemma shows asymptotic results on the estimator $\hat{\theta}$ minimize the loss function $\ell(., X, Y)$ over an iid dataset. Further, we assume that this dataset is generated using a model parametrized by $\theta_0$. i.e. $\mathbb{E}\left[\frac{\partial}{\partial\theta}\ell(\theta_0, X, Y)\right] = 0$. Further, we often overload $||.||_2$ with $L_2(\mathcal{D})$ norm when operated on a function formally, $||t||_2 = \sqrt{\mathbb{E}_X\left[(t(X))^2\right]}$.

**Lemma B.1.** *Define the IF$(\theta, X, Y) := -\left(\mathbb{E}\left[\frac{\partial^2}{\partial\theta^2}\ell(\theta, X, Y)\right]\right)^{-1}\frac{\partial}{\partial\theta}\ell(\theta, X, Y)$. Let the estimator $\hat{\theta}_n$ minimize the loss function $\ell(.)$ over the dataset $\{X_i, Y_i\}_{i=1}^n$. Under standard regularity assumptions,*

$$\sqrt{n}(\hat{\theta}_n - \theta_0) \xrightarrow{d} \mathcal{N}(0, \mathit{Var}(\mathit{IF}(\theta_0, X, Y)))$$

We now compute the influence function for logistic losses $\ell(\theta, X, Y) := \log(\sigma(2aY\langle\theta, X\rangle))$. Now, observe that $\frac{\partial}{\partial\theta}\ell(\theta, X, Y) = 2\sigma(-2aY\langle\theta, X\rangle)XY$. Further, $\frac{\partial^2}{\partial\theta^2}\ell(\theta, X, Y) = 4a^2\sigma(2aY\langle\theta, X\rangle)\sigma(-2aY\langle\theta, X\rangle)XX^T$. This argument is fairly standard and we restate the derivation below for expository purposes.

Since $\mathbb{E}\left[\frac{\partial}{\partial\theta}\ell(\theta_0, X, Y)\right] = 0$, the variance of $\frac{\partial}{\partial\theta}\ell(\theta_0, X, Y)$ equals $\mathbb{E}\left[\frac{\partial}{\partial\theta_0}\ell(\theta_0, X, Y)\frac{\partial}{\partial\theta_0}\ell(\theta_0, X, Y)^T\right]$.

Now,

$$\mathrm{Var}\left(\frac{\partial}{\partial\theta_0}\ell(\theta_0, X, Y)\right) = 4a^2\mathbb{E}[\sigma(-2aY\langle\theta_0, X\rangle)\sigma(-2aY\langle\theta_0, X\rangle)XX^T]$$

$$\overset{(a)}{=} 4a^2\mathbb{E}[\sigma(-2a\langle\theta_0, X\rangle)\sigma(-2a\langle\theta_0, X\rangle)\sigma(2a\langle\theta_0, X\rangle) \tag{15}$$

$$+ \sigma(2a\langle\theta_0, X\rangle)\sigma(2a\langle\theta_0, X\rangle)\sigma(-2a\langle\theta_0, X\rangle)XX^T] \tag{16}$$

$$= 4a^2\mathbb{E}\left[\sigma(-2a\langle\theta_0, X\rangle)\sigma(2a\langle\theta_0, X\rangle)XX^T\right] \tag{17}$$

$(a)$ follows via conditioning on $X$ and the fact that $\mathbb{P}(Y = 1 \mid X) = \sigma(2aY\langle\theta_o, X\rangle)$. $(b)$ follows from the fact that $\sigma(r) + \sigma(-r) = 1$

Using similar arguments, one can show that

$$\mathbb{E}\left[\frac{\partial^2}{\partial\theta^2}\ell(\theta_0, X, Y)\right] = 4a^2\mathbb{E}\left[\sigma(-2a\langle\theta_0, X\rangle)\sigma(2a\langle\theta_0, X\rangle)XX^T\right] \tag{18}$$

Since $\mathrm{Var}(\mathrm{IF}(\theta_0, X, Y)) = \left(\mathbb{E}\left[\frac{\partial^2}{\partial\theta^2}\ell(\theta, X, Y)\right]\right)^{-2}\mathrm{Var}\left(\frac{\partial}{\partial\theta_0}\ell(\theta_0, X, Y)\right)$, applying Lemma B.1, we have

**Lemma B.2.** $\sqrt{n}(\hat{\theta}_{CH} - \theta_0) \xrightarrow{d} \mathcal{N}(0, \left(4a^2\mathbb{E}\left[\sigma(-2a\langle\theta_0, X\rangle)\sigma(2a\langle\theta_0, X\rangle)XX^T\right]\right)^{-1})$ *where the estimator $\hat{\theta}_{CH}$ denotes the estimator obtained from logistic regression on the preference data.*

Since $\sigma(r)\sigma(-r)$ scales asymptotically as $\exp(-|r|)$, one can above that the variance term can scale exponentially in $S$ asymptotically for larger norms of $\theta_0$ where $S$ denotes the $\ell_2$ bound on $\theta_0$.

---

[5]The constant $2a$ follows from the probability $P(Y \mid X)$ in (1).

## B.2 Computing Rates for Orthogonal Loss (6) and (12)

We first restate the notations from Appendix B. In this section, we consider the linear class $\mathcal{R} = \{x \to \langle x, \theta \rangle : \theta \in \mathbb{R}^d\}$ so that $r_\theta(X) = \langle \theta, X \rangle$ and the true reward model $r_o(X) = \langle \theta_o, X \rangle$ and the goal is to estimate $\theta_o$. We further assume that the $\ell_2$ norm of queries $x$ satisfies $||x||_2 \leq 1$. Further recall that $Z$ denotes the tuple $(X, Y, T)$.

Recall from Section 4 that the asymptotic rates for linear function classes holds under cross-fitting and data-splitting. We now restate cross-fitting from [CNR18] for expository purposes which involves training nuisances out-of-fold and evaluating on each held-out fold before aggregating. Further we denote $\ell^{\text{ortho}}$ and $\ell^{\text{ortho}-2}$ as the point-wise versions of orthogonal losses $\mathcal{L}^{\text{ortho}}$ and $\mathcal{L}^{\text{ortho}-2}$.

While we state cross-fitting under the orthogonal loss $\mathcal{L}^{\text{ortho}}$, one can easily state it for the orthogonal loss $\mathcal{L}^{\text{ortho}-2}$ with the only difference in the nuisance functions computed.

---

**Input:** $\mathcal{S} = \{(X_i^{(1)}, X_i^{(2)}, Y_i, T_i)\}_{i=1}^n$
**Goal:** Estimate reward model $r(\cdot)$
 1: Partition the training data into $B$ equally sized folds (call it $\{\mathcal{S}_p\}_{p=1}^B$).
 2: For each fold $\mathcal{S}_i$, estimate the nuisance reward function $\hat{\mathfrak{r}}(\cdot)^{(p)}$ and time function $\hat{t}(\cdot)^{(p)}$ using the out of fold data points.
 3: Use $\hat{\mathfrak{r}}(\cdot)$ and $\hat{t}(\cdot)$ as nuisance to estimate the reward model $r(\cdot)$ using an orthogonal loss by minimising $\frac{1}{n} \sum_{p=1}^B \sum_{i \in \mathcal{S}_p} \ell(r; \hat{t}^{(p)}, \hat{\mathfrak{r}}^{(p)}; Z_i)$ over observed data points $Z_i = (X_i, Y_i, T_i)$ for orthogonal loss $\ell = \ell^{\text{ortho}}$

---

Meta-Algorithm 2: Estimate Reward Model via Orthogonal Loss and cross-fitting

Under data-reuse the nuisance is fitted on one half of the data and the orthogonalized loss is minimized on the other.

Before we present the asymptotic rates for linear reward classes for orthogonal losses, we first state the following theorem from [CNR18, Section 3.2]. Recall that $g(\cdot)$ is the nuisance function, given by the tuple $(y(\cdot), t(\cdot))$ under the orthogonal loss $\mathcal{L}^{\text{ortho}-2}$, and by the tuple $(\mathfrak{r}(\cdot), t(\cdot))$ under the orthogonal loss $\mathcal{L}^{\text{ortho}}$.

### B.2.1 Moment function and Neyman orthogonality setup from [CNR18, Section 3]

Further, following the notation in Section 5, we define the nuisance function class by $\mathcal{G}$ and we also assume that the nuisance estimates also belong to this class.

Define the *sample moment function*

$$m(\theta, g, Z) = j(g, Z)\theta + v(g, Z),$$

and the associated *population moments*

$$M(\theta, g) = \mathbb{E}[m(\theta, g, Z)], \qquad J(g) = \mathbb{E}[j(g, Z)], \qquad V(g) = \mathbb{E}[v(g, Z)].$$

Let $\theta_o$ be the (unique) parameter solving the population moment condition

$$M(\theta_o, g_o) = 0,$$

where $g_o$ denotes the true nuisance function.

We refer to $j(g, Z)$ as the *sample Jacobian matrix* and to its expectation $J(g)$ as the *population Jacobian matrix*.

To map to our set up where we minimize the pointwise loss $\ell^{\text{ortho}}(\theta, g, Z)$ or $\ell^{\text{ortho}-2}(\theta, g, Z)$, we can compute the function $m(.)$ by taking the gradient with respect to the first component of the loss function.

We now state assumption 3.1 from [CNR18]. Because we focus on the moment function $m(\cdot)$, Neyman orthogonality is defined solely with respect to the nuisance function, whereas for a loss function $\ell(.)$ it is defined with respect to both the nuisance function and the target parameter.

**Assumption B.3** (Neyman Orthogonality and continuity). The directional derivative $D_g M(\theta_o, g_o)[g-g_o]$ equals zero (satisfying Neymann orthogonality) and further, $D_{gg} M(\theta, g)[g-g_o]$ is continuous in a small neighborhood of $g_o$. Further, the eigen values of the Jacobian matrix $J(g_o)$ lie between constants $c_1$ and $c_2$.

We now let $c_0 > 0$, $c_1 > 0$, $s > 0$ and $q > 2$ be some finite constants such that $c_0 \leq c_1$; and let $\delta_n$ be some sequences of positive constants converging to zero such that $\delta_n \geq n^{-1/2}$. Recall that $\hat{g}$ denotes the nuisance parameters estimated from the first stage.

**Assumption B.4** (Score Regularity and Quality of Nuisance Estimates). The following moment conditions hold:

- The $q^{\text{th}}$ order moment conditions hold i.e. $\sup_{g \in \mathcal{G}} \left( \mathbb{E}\left[ ||m(\theta_o, g, Z)||^q \right]^{1/q} \right)$ and $\sup_{g \in \mathcal{G}} \left( \mathbb{E}\left[ ||j(g, Z)||^q \right]^{1/q} \right)$ are bounded by constant $c$.

- The Jacobian $J(\cdot)$ satisfies $||J(\hat{g}) - J(g_o)||_{\text{op}}^2 \leq \delta_n$ where $||.||_{\text{op}}$ denotes the operator norm.

- The moment function satisfies $\left( ||m(\theta_o, \hat{g}, Z) - m(\theta_o, g_o, Z)||_2^2 \right)^{1/2} \leq \delta_n$.

- The second order directional derivative in direction of the nuisance error converges to zero faster than $n^{-1/2}$ i.e. $\sqrt{n} D_{gg} M(\theta_o, \bar{g})[\hat{g} - g_o] = o_p(1)$, $\forall \bar{g} = \tau \hat{g} + (1-\tau) g_o, \tau \in [0, 1]$.

- The variance $\mathbb{E}\left[ m(\theta_o, g_o, Z) m(\theta_o, g_o, Z)^\top \right]$ is lower bounded.

Under these Assumption B.3 and Assumption B.4 the following lemma holds on the estimate $\hat{\theta}$ under data-splitting and cross-fitting.

**Lemma B.5.** *Under these Assumption B.3 and Assumption B.4, the estimate $\hat{\theta}$ is asymptotically linear under data-splitting and cross-fitting. :*

$$\sqrt{n}\left( \hat{\theta} - \theta_0 \right) = -\frac{1}{\sqrt{n}} \sum_{i=1}^n J(g_0)^{-1} m(\theta_0, g_0, Z) + o_p(1),$$

*This implies that the following convergence holds in distribution* $\sqrt{n}(\hat{\theta} - \theta_0) \xrightarrow{d} \mathcal{N}(0, J(g_0)^{-2} \mathbb{E}[m(\theta_0, g_0; Z) m(\theta_0, g_0; Z)^\top])$

With this setup, we now compute the guarantees for orthogonal loss $\mathcal{L}^{\text{ortho}}$ and $\mathcal{L}^{\text{ortho}-2}$.

## B.3 Asymptotic Rates for orthogonal loss $\mathcal{L}^{\text{ortho}}$

In this notation $\ell(.)$ refers to the pointwise version of orthogonalized loss $\mathcal{L}^{\text{ortho}}$. Recall that the nuisance function $g(\cdot)$ denotes the tuple of nuisance functions $(\mathfrak{r}, t)$ with $\hat{g}_o(\cdot) = (r_o, t_o)$ denoting their true values. Further the nuisance function $\hat{g} = (\hat{\mathfrak{r}}, \hat{t})$ denotes an estimate of true nuisance function $\hat{g}_o(\cdot) = (r_o, t_o)$ after the first stage.

We now adopt the following notation. Recall that $\theta_o \in \mathbb{R}^d$ denoted the true parameter vector. We define the moment function by

$$m(\theta, y, t; X, Y, T) := \frac{1}{2} \nabla_\theta \ell(\theta, r; X, Y, T),$$

which, after algebraic manipulation, can be written as

$$m(\theta, \mathfrak{r}, t; X, Y, T) = \frac{1}{a} \left( \frac{\langle X, \theta \rangle}{a} t(X)^2 + \frac{\mathfrak{r}(X) t(X)}{a} (T - t(X)) - Y t(X) \right) X.$$

In addition, the two auxiliary functions can be defined as:

$$j(t; X) := \frac{t(X)^2}{a^2} X X^T \text{ and } v(\mathfrak{r}, t; X, Y, T) := (1/a)\left( -Y t(X) + \frac{\mathfrak{r}(X) t(X)}{a} (T - t(X)) \right) X \quad (19)$$

We also compute the expectation-based mappings:

$$M(\theta, \mathfrak{r}, t) := \mathbb{E}\Big[m(\theta, \mathfrak{r}, t; X, Y, T)\Big], \qquad J(t) := a^{-2}\mathbb{E}\Big[t(X)^2 XX^T\Big], \tag{20}$$

$$V(\mathfrak{r}, t) := a^{-1}\mathbb{E}\Big[\Big((t_o(X) - t(X))\frac{\mathfrak{r}(X)t(X)}{a} - y_o(X)\,t(X)\Big)X\Big]. \tag{21}$$

A straightforward calculation shows that

$$M(\theta, \mathfrak{r}, t) = J(t)\,\theta + V(\mathfrak{r}, t).$$

**Claim B.6** (Orthogonality of the loss and continuity (assumption B.3)). $D_g M(\theta_o, g_0)[\hat{g} - g_0] = 0$ and $D_{gg}M(\theta, g)[g - g_o]$ is continuous in $\mathcal{G}$

*Proof.*

$$D_g m(\theta_0, g_0, X, Y, T)[\hat{g} - g_0] = 2a^{-2}X\langle X, \theta_0\rangle t_0(X)(\hat{t}(X) - t_0(X)) - 2a^{-2}Xr_0(X)t_o(X)(\hat{t}(X) - t_0(X))$$
$$- a^{-1}X(Y - a^{-1}Tr_o(X))(\hat{t}(X) - t_0(X)) + a^{-2}t_o(X)(T - t_o(X))(\mathfrak{r}(X) - r_o(X))$$

Since, $r_o(X) = \langle X, \theta_o\rangle$ and the fact that $ay_o(X) = r_o(X)t_o(X)$, we obtain that $\mathbb{E}\left[D_g m(\theta_0, g_0, X, Y, T)[\hat{g} - g_0] \mid X\right] = 0$

The continuity of the second order functional derivative naturally follows.

$\square$

Recall that we have the following assumption of invertibility of $J(t_o) = \mathbb{E}\left[t_o(X)^2 XX^\top\right]$ from Theorem 4.1.

**Assumption B.7** (Invertibility of the Jacobian). We assume that the Jacobian matrix satisfies

$$\|J(t_o)^{-1}\|_{\text{op}} = \left\|\mathbb{E}\left[-a^{-2}t_o(X)^2\,XX^T\right]^{-1}\right\|_{\text{op}} \leq C,$$

for some constant $C > 0$. This claim is justified under mild conditions on the boundedness of the reward and eigen values of the data matrix $\mathbb{E}[XX^T]$.

We now prove the following claim on nuisance estimation.

**Claim B.8** (Nuisance Estimation Rates). *There exists a black-box learner such that its root mean squared error (RMSE) satisfies*

$$\|\hat{t} - t_o\|_{L_2(\mathcal{D})} = o\Big(\frac{1}{n^{\beta_1}}\Big) \quad \text{with } \beta_1 \geq \frac{1}{4},$$

*and*

$$\|\hat{\mathfrak{r}} - r_o\|_{L_2(\mathcal{D})} = o\Big(\frac{1}{n^{\beta_2}}\Big) \quad \text{with } \beta_2 \geq \frac{1}{2} - \beta_1.$$

*These conditions ensure that the nuisance estimates converge sufficiently fast as the sample size $n$ increases.*

*Proof.* We show that these slow rates can be achieved under both conditions of plug-in where $\hat{t}(\cdot) = \frac{\tanh \hat{\mathfrak{r}}(\cdot)}{\mathfrak{r}(\cdot)}$ with $\hat{\mathfrak{r}}(\cdot)$ estimated via log-loss. Alternately, one could separately estimate $t(\cdot)$ as well to obtain these slow rates. A discussion is presented in Appendix B.5.

$\square$

Furthermore, we assert the following claim regarding the boundedness of the nuisance function $t_o(X)$.

**Claim B.9.**

$$t_o(X) = a\frac{\tanh\big(a\langle\theta_o, X\rangle\big)}{\langle\theta_o, X\rangle} \leq a^2.$$

A brief inspection using the properties of the hyperbolic tangent function (notably, that $\tanh(x) \leq x$ for $x \geq 0$) confirms this bound.

We now show that assumptions in Assumption B.4 hold for our loss function using the following three claims and lemmas namely Claim B.10, Claim B.11, Lemma B.12 and Claim B.13

**Claim B.10.** *The jacobian $J(\cdot)$ satisfies $J(\hat{t}) - J(t_o) = o(1)$ and the $q^{th}$ order moment conditions hold i.e. i.e. $\sup_{g \in \mathcal{G}} \left( \mathbb{E}\left[ ||m(\theta_o, g, Z)||^q \right]^{1/q} \right)$ and $\sup_{g \in \mathcal{G}} \left( \mathbb{E}\left[ ||j(g, Z)||^q \right]^{1/q} \right)$ is bounded.*

*Proof.* Write

$$J(t) - J(t_o) = a^{-2} \mathbb{E}\left[ \left( t(X)^2 - t_o(X)^2 \right) X X^T \right].$$

Since the operator norm is bounded by the Frobenius norm, we have

$$\|J(t) - J(t_o)\|_{op} \leq \|J(t) - J(t_o)\|_F.$$

Using the Frobenius norm, we estimate

$$\|J(t) - J(t_o)\|_F = a^{-2} \left\| \mathbb{E}\left[ \left( t(X)^2 - t_o(X)^2 \right) X X^T \right] \right\|_F.$$

Using the boundedness of $X$ (i.e. $\|X\| \leq 1$), we have

$$\|X X^T\|_F = \|X\|^2 \leq 1.$$

Notice that

$$|t(X)^2 - t_o(X)^2| = |t(X) - t_o(X)| \, |t(X) + t_o(X)|.$$

Both $t$ and $t_o$ are uniformly bounded ($|t(X)|, |t_o(X)| \leq a^2$), then $|t(X) + t_o(X)| \leq 2$. Hence, $|t(X)^2 - t_o(X)^2| \leq 2a^2 \, |t(X) - t_o(X)|$. Thus, we obtain

$$\|J(t) - J(t_o)\|_F \leq 2a^{-2} \mathbb{E}\left[ 2 \, |t(X) - t_o(X)| \cdot \right] = 2a^{-2} \mathbb{E}\left[ |t(X) - t_o(X)| \right].$$

Finally, applying Jensen's inequality (or noting that $\mathbb{E}[|t(X) - t_o(X)|] \leq \|t - t_o\|_{L_2(\mathcal{D})}$) yields

$$\|J(t) - J(t_o)\|_{op} \leq 2a^{-2} \|t - t_o\|_{L_2(\mathcal{D})}.$$

This concludes the proof as the nuisance estimators are consistent i.e $||\hat{g} - g_o||_{L_2(\mathcal{D})} = o_p(1)$

We now check the moment condition. This naturally follows from the fact that $\mathbb{E}[T^\alpha \mid X]$ is bounded for every $\alpha \geq 1$ and the fact that rest all random variables are functions are bounded. $\qquad\square$

**Claim B.11.** *Let $g$ be a vector-valued function defined as*

$$\hat{g}(x) = \begin{bmatrix} \hat{\mathfrak{r}}(x) \\ \hat{t}(x) \end{bmatrix}.$$

*Then, for any $\bar{g} = \tau \hat{g} + (1 - \tau) g_o$ with $\tau \in [0, 1]$, we have*

$$\sqrt{n} \, D_{gg} M\left( \theta_o, \bar{g} \right)[\hat{g} - g_o] = o_p(1).$$

*Proof.* We wish to control the second-order term in the expansion of $M(\theta_o, g)$ around $g_o$. By Taylor's theorem in the direction $\hat{g} - g_o$, we have

$$D_{gg} M\left( \theta_o, \bar{g} \right)[\hat{g} - g_o] = \frac{\partial^2}{\partial s^2} M\left( \theta_o, \, \bar{g} + s \left( \hat{g} - g_o \right) \right)\Big|_{s=0}.$$

This term decomposes into two parts:

$$D_{gg} M\left( \theta_o, \bar{g} \right)[\hat{g} - g_o] = \underbrace{\frac{\partial^2}{\partial s^2} \left( J(\bar{t} + s \left( \hat{t} - t_o \right)) \theta_o \right)\Big|_{s=0}}_{(I)} + \underbrace{\frac{\partial^2}{\partial s^2} V\left( \bar{t} + s \left( \hat{t} - t_o \right), \, \bar{\mathfrak{r}} + s \left( \hat{\mathfrak{r}} - r_o \right) \right)\Big|_{s=0}}_{II}.$$

$$\tag{22}$$

First Term (I): For $J(t) = a^{-2}\mathbb{E}[t(X)^2 XX^T]$, we have

$$\frac{\partial^2}{\partial s^2}\left(\left(\bar{t}(X) + s\left(t(X) - t_o(X)\right)\right)^2\right)\Big|_{s=0} = 2\left(t(X) - t_o(X)\right)^2.$$

Thus,

$$\frac{\partial^2}{\partial s^2}\left(J(\bar{t} + s\,(\hat{t} - t_o))\theta_o\right)\Big|_{s=0} = 2a^{-2}\,\mathbb{E}\left[(\hat{t}(X) - t_o(X))^2\,XX^T\right]\theta_o.$$

Since $\|XX^T\|_{\mathrm{op}}$ is bounded (using $\|X\| \leq 1$) and $\theta_o$ is fixed, it follows that

$$\left\|2\,\mathbb{E}\left[(\hat{t}(X) - t_o(X))^2\,XX^T\right]\theta_o\right\| = O\left(\|\hat{t} - t_o\|_2^2\right).$$

Under Claim B.8, we have $\|t - t_o\|_2^2 = o\left(\frac{1}{\sqrt{n}}\right)$, so that

$$\sqrt{n}\,O\left(\|\hat{t} - t_o\|_2^2\right) = o(1).$$

Second Term (II): For the function $V(t, y)$, we have

$$\frac{\partial^2}{\partial s^2}V\left(\bar{t} + s\,(\hat{t} - t_o),\ \bar{y} + s\,(\hat{y} - y_o)\right)\Big|_{s=0} = a^{-2}\mathbb{E}\left[(2T - 4t(X))(\hat{r}(X) - r_o(X))(\hat{t}(X) - t_o(X))\,X\right]$$

$$- \frac{4}{a^2}\mathbb{E}\left[\mathfrak{r}(X)(\hat{t} - t_o(X))^2 X\right]$$

$$= a^{-2}\mathbb{E}\left[(t_o(X) - 2t(X))(\hat{r}(X) - r_o(X))(\hat{t}(X) - t_o(X))\,X\right]$$

$$- \frac{4}{a^2}\mathbb{E}\left[\mathfrak{r}(X)(\hat{t}(X) - t_o(X))^2 X\right].$$

$$(23)$$

The last equality follows via conditioning on the first term. Using the boundedness of $X, t(X)$ and $\mathfrak{r}(X)$ and applying the Cauchy–Schwarz inequality, we obtain

$$\left\|\mathbb{E}\left[(t_o(X) - 2t(X))(\hat{t}(X) - t_o(X))\,(\hat{r}(X) - r_o(X))\,X\right]\right\| \leq C\,\|\hat{t} - t_o\|_{L_2(\mathcal{D})}\,\|\hat{r} - r_o\|_{L_2(\mathcal{D})}, \tag{24}$$

$$\left\|\mathbb{E}\left[\mathfrak{r}(X)(\hat{t}(X) - t_o(X))^2\right]\right\| \leq C\|\hat{t} - t_o\|_{L_2(\mathcal{D})}^2 \tag{25}$$

for some constant $C > 0$. By Claim B.8, the product $\|\hat{t} - t_o\|_{L_2(\mathcal{D})}\,\|\hat{r} - r_o\|_{L_2(\mathcal{D})}$ is $o\left(\frac{1}{\sqrt{n}}\right)$ and $\|\hat{t} - t_o\|_{L_2(\mathcal{D})}^2$ is $o(1/n)$. Therefore,

$$\sqrt{n}\left\|\mathbb{E}\left[(\hat{t}(X) - t_o(X))\,(\hat{r}(X) - r_o(X))\,X\right]\right\| = o(1).$$

Combining the two terms, we conclude that

$$\sqrt{n}\,D_{gg}M\left(\theta_o, \bar{g}\right)[\hat{g} - g_o] = o_p(1).$$

This completes the proof. $\qquad\square$

**Lemma B.12.** *The following holds*

$$\mathbb{E}[\|m(\theta_o, \mathfrak{r}, t; X, Y, T) - m(\theta_o, r_o, t_o; X, Y, T)\|^2] = O\left(\|t - t_o\|^2_{L_2(\mathcal{D})} + \|\mathfrak{r} - r_o\|^2_{L_2(\mathcal{D})}\right)$$

*Given that* $\|\hat{t} - t_o\|_{L_2(\mathcal{D})} = o(1)$ *and* $\|\hat{r} - r_o\|_{L_2(\mathcal{D})} = o(1)$, *we have* $\mathbb{E}[\|m(\theta_o, \hat{g}, Z) - m(\theta_o, g_0; Z)\|^2] = o(1)$

*Proof.* We have

$$m(\theta_o, \mathfrak{r}, t; X, Y, T) - m(\theta_o, r_o, t_o; X, Y, T) \tag{26}$$

$$= a^{-2}\left(\langle\theta_o, X\rangle(t(X)^2 - t_o(X)^2) + (\mathfrak{r}(X)t(X) - r_o(X)t_o(X))T + (r_0(X)t_o(X)^2 - \mathfrak{r}(X)t(X)^2) + aY(t_o(X) - t(X))\right)X \tag{27}$$

$$(\mathfrak{r}(X)t(X) - r_o(X)t_o(X))T + (r_0(X)t_o(X)^2 - \mathfrak{r}(X)t(X)^2)$$
$$=(\mathfrak{r}(X)t(X) - r_o(X)t_o(X))(T - t_o(X)) + \mathfrak{r}(X)t(X)(t_o(X) - t(X))$$
$$=(\mathfrak{r}(X)(t(X) - t_o(X)) - t_o(X)(\mathfrak{r}(X) - r_o(X)))(T - t_o(X)) + \mathfrak{r}(X)t(X)(t_o(X) - t(X))$$
$$=\mathfrak{r}(X)(t(X) - t_o(X))(T - t_o(X)) - t_o(X)(\mathfrak{r}(X) - r_o(X))(T - t_o(X)) + \mathfrak{r}(X)t(X)(t_o(X) - t(X))$$

Using the fact that $||X||_2 \leq 1$, we have

$$\mathbb{E}\Big(m(\theta_o, \mathfrak{r}, t; X, Y, T) - m(\theta_o, r_o, t_o; X, Y, T)\Big)_i^2 = \sqrt{d}a^{-2}\mathbb{E}\Bigg[\Big(\langle\theta_o, X\rangle(t(X) - t_o(X))(t(X) + t_o(X))$$

$$+ \mathfrak{r}(X)(t(X) - t_o(X))(T - t_o(X)) - t_o(X)(\mathfrak{r}(X) - r_o(X))(T - t_o(X)) + \mathfrak{r}(X)t(X)(t_o(X) - t(X)) + aY(t_o(X) - t(X))\Big)^2\Bigg]$$

$$(28)$$

The desired result is immediate via the the identity that $(a+b+c+d+e)^2 \leq 5(a^2+b^2+c^2+d^2+e^2)$ except for the following terms

$$\mathbb{E}\left[\mathfrak{r}(X)^2(t(X) - t_o(X))^2(T - t_o(X))^2\right] = \mathbb{E}\left[\mathbb{E}\left[\mathfrak{r}(X)^2(t(X) - t_o(X))^2(T - t_o(X))^2 \mid X\right]\right]$$
$$= \mathbb{E}\left[\mathfrak{r}(X)^2(t(X) - t_o(X))^2\mathrm{Var}(T \mid X)\right] \leq C||t - t_o||^2_{L_2(\mathcal{D})}$$

for some constant $C$.

One can similarly argue that

$$\mathbb{E}\left[t_o(X)^2(\mathfrak{r}(X) - r_o(X))^2(T - t_o(X))^2\right] \leq C|\mathfrak{r} - r_o|^2_{L_2(\mathcal{D})}$$

Finally this gives us

$$\mathbb{E}\Big(m(\theta_o, \mathfrak{r}, t; X, Y, T) - m(\theta_o, r_o, t_o; X, Y, T)\Big)_i^2 \leq C\left(||t - t_o||^2_{L_2(\mathcal{D})} + (4S + 1)||\mathfrak{r} - r_o||^2_{L_2(\mathcal{D})}\right)$$

$$\square$$

**Claim B.13.** *Each component of the matrix $j(t_o, X)$ satisfies*

$$\mathbb{E}\Big[j_{pq}(t_o, X)^2\Big] = O(1).$$

*Furthermore, the components of the moment function satisfy*

$$\mathbb{E}\Big[m_q(\theta_o, r_o, t_o)^2\Big] = O(1).$$

*Proof.* The first statement follows directly from the boundedness of $t_o(X)$ and the boundedness of $X$. For the second statement, note that

$$\mathbb{E}\Big[m_q(\theta_o, y_o, t_o)^2\Big] \leq 2a^{-2}\left(\mathbb{E}\Big[t_o(X)^2\Big] + \mathbb{E}\Big[a^{-2}\langle X, \theta_o\rangle^2 t_o(X)^2\Big] \cdot \mathbb{E}\Big[y_o(X)\big(T - t_o(X)\big)^2\Big]\right).$$

Since the functions $r_o$ and $t_o$ are bounded and the variance of $T$ is also bounded, it follows that

$$\mathbb{E}\Big[m_q(\theta_o, r_o, t_o)^2\Big] = O(1).$$

$$\square$$

Invoking Lemma B.5, we conclude that the estimator $\hat{\theta}$ obtained via minimising orthogonal loss $\mathcal{L}^{\mathrm{ortho}}$ is asymptotically linear. In particular,

$$\sqrt{n}\left(\hat{\theta} - \theta_o\right) = -\frac{1}{\sqrt{n}}\sum_{i=1}^{n} IF(\theta_o, r_o, t_o; X_i, Y_i, T_i) + o_p(1), \qquad (29)$$

where the influence function is given by

$$IF(\theta_o, r_o, t_o; X, Y, T) = \mathbb{E}[a^{-2}t_o(X)^2 XX^T]^{-1} m(\theta_o, r_o, t_o; X, Y, T)$$

Note that by the definition of $\theta_o$ we have $\mathbb{E}\left[m(\theta_o, y_o, t_o; X, Y, T)\right] = 0$. Now

$$m(\theta_o, r_o, t_o; X, Y, T) = \frac{1}{a}\Big(\frac{\langle X, \theta\rangle}{a}\, t_o(X)^2 + y_o(X)\big(T - t_o(X)\big) - Y\, t_o(X)\Big)X \qquad (30)$$

$$= \frac{1}{a}\left(y_o(X)\big(T - t_o(X)\big) - t_o(X)\big(Y - y_o(X)\big)\right) \qquad (31)$$

Thus using the fact that $\mathbb{E}[T \mid X] = t_o(X), \mathbb{E}[Y \mid X] = y_o(X)$ and $\mathrm{Var}(Y \mid X) = 1 - y_o(X)^2$, we get

$$\mathrm{Covar}(m(\theta_o, r_o, t_o; X, Y, T)) = \frac{1}{a^2}\mathbb{E}\left[\Big(t_o(X)^2(1 - y_o(X)^2) + \mathrm{Var}(T|X)\cdot y_o(X)^2\Big)XX^T\right]$$

$$= \frac{1}{a^4}\mathbb{E}\left[t_o(X)^3 XX^T\right]$$

The last step follows from the fact that (Claim B.24). Thus, the covariance of the influence function is given by

$$\mathrm{Covar}(IF) = \mathbb{E}[t_o(X)^2 XX^T]^{-2}\mathbb{E}[t_o(X)^3 XX^\top]$$

In particular, this concludes the proof of Theorem 4.1. We restate it below for clarity.

**Theorem.** *Let $\hat{\theta}_{\mathrm{ortho}}$ minimize the orthogonal loss $\mathcal{L}^{\mathrm{ortho}}$. If $\mathbb{E}[\,t_0(X)^2 XX^\top\,]$ is invertible, then*

$$\sqrt{n}(\hat{\theta}_{\mathrm{ortho}} - \theta_o) \xrightarrow{d} \mathcal{N}(0, \Sigma)$$

*where*

$$\Sigma = \mathbb{E}\left[\left(\frac{a\tanh(a\langle\theta_o, X\rangle)}{\langle\theta_o, X\rangle}\right)^2 XX^\top\right]^{-2} \mathbb{E}\left[\left(\frac{a\tanh(a\langle\theta_o, X\rangle)}{\langle\theta_o, X\rangle}\right)^3 XX^\top\right].$$

From claim Claim B.25 and the fact that $\frac{\tanh x}{x} \leq 1$, one can observe that this variance $\Sigma$ is smaller than the variance computed in Lemma B.2.

## B.4 Guarantees for Orthogonal Loss $\mathcal{L}^{\mathrm{ortho}-2}$

In this section, we prove the following theorem which is the analog of Theorem 4.1 while minimising the orthogonal loss $\mathcal{L}^{\mathrm{ortho}-2}$. We obtain identical asymptotic rates as obtained in $\mathcal{L}^{\mathrm{ortho}}$.

**Theorem B.14.** *Let $\hat{\theta}_{\mathrm{ortho}}$ minimize the orthogonal loss $\mathcal{L}^{\mathrm{ortho}-2}$. If $\mathbb{E}[\,t_0(X)^2 XX^\top\,]$ is invertible, then*

$$\sqrt{n}(\hat{\theta}_{\mathrm{ortho}} - \theta_o) \xrightarrow{d} \mathcal{N}(0, \Sigma)$$

*where*

$$\Sigma = \mathbb{E}\left[\left(\frac{a\tanh(a\langle\theta_o, X\rangle)}{\langle\theta_o, X\rangle}\right)^2 XX^\top\right]^{-2} \mathbb{E}\left[\left(\frac{a\tanh(a\langle\theta_o, X\rangle)}{\langle\theta_o, X\rangle}\right)^3 XX^\top\right].$$

We now define the moment functions as defined in Appendix B.2.1. Further in this setup, we use $\ell(\cdot)$ to denote the pointwise version of orthogonal loss $\mathcal{L}^{\mathrm{ortho}-2}$. Recall that the nuisance function $g(\cdot)$ denotes the tuple of nuisance functions $(y, t)$ with $\hat{g}_o(\cdot) = (y_o, t_o)$ denoting their true values. Further the nuisance function $\hat{g} = (\hat{y}, \hat{t})$ denotes an estimate of true nuisance function $\hat{g}_o(\cdot) = (y_o, t_o)$ after the first stage. Let $\theta_o \in \mathbb{R}^d$ denote the true parameter vector. We define the moment function by

$$m(\theta, y, t; X, Y, T) := \frac{1}{2}\nabla_\theta \ell(\theta, r; X, Y, T),$$

which, after algebraic manipulation, can be written as

$$m(\theta, y, t; X, Y, T) = \frac{1}{a}\Big(\frac{\langle X, \theta\rangle}{a}\, t(X)^2 + y(X)\big(T - t(X)\big) - Y\, t(X)\Big)X.$$

In addition, we introduce the auxiliary functions:

$$j(t; X) := \frac{t(X)^2}{a^2}\, XX^T \quad \text{and} \quad v(y, t; X, Y, T) := (1/a)\Big(-Y\, t(X) + y(X)\big(T - t(X)\big)\Big)X \quad (32)$$

We also define the expectation-based mappings:

$$M(\theta, y, t) := \mathbb{E}\Big[m(\theta, y, t; X, Y, T)\Big], \qquad J(t) := a^{-2}\mathbb{E}\Big[t(X)^2\, XX^T\Big],$$

$$V(y, t) := a^{-1}\mathbb{E}\Big[\big((t_o(X) - t(X))y(X) - y_o(X)\, t(X)\big)X\Big].$$

A straightforward calculation shows that

$$M(\theta, y, t) = J(t)\,\theta + V(y, t).$$

**Claim B.15** (Orthogonality of the loss and continuity (assumption B.3)). $D_g M(\theta_o, g_0)[\hat{g} - g_0] = 0$ and $D_{gg}M(\theta, g)[g - g_o]$ is continuous in $\mathcal{G}$

*Proof.*

$$D_g m(\theta_0, g_0, X, Y, T)[\hat{g} - g_0] = 2a^{-2}X\langle X, \theta_0\rangle t_0(X)(\hat{t}(X) - t_0(X)) - a^{-1}Xy_0(X)(\hat{t}(X) - t_0(X))$$
$$- a^{-1}YX(\hat{t}(X) - t_0(X)) + a^{-1}(T - t_0(X))(\hat{y}(X) - y_0(X))$$

$$D_g M(\theta_0, g_0)[\hat{g} - g_0] = \mathbb{E}[D_t m(\theta_0, t_0, y_0, X, Y, T)[\hat{t} - t_0, \hat{y} - y_0]]$$
$$= (1/a)\mathbb{E}\big[\mathbb{E}\big[X(2a^{-1}\langle X, \theta_0\rangle t_0(X) - y_0(X) - Y)(\hat{t}(X) - t_0(X))|X\big]\big]$$
$$+ \mathbb{E}[\mathbb{E}[(T - t_0(X))(\hat{y}(X) - y_0(X))|X]]$$

Note $\mathbb{E}[(T - t_0(X))(\hat{y}(X) - y_0(X))|X] = 0$ since $\mathbb{E}[T|X] = t_0(X)$.

Also, observe that $\mathbb{E}[2\langle X, \theta_0\rangle t_0(X) - y_0(X) - Y|X] = 0$ since $\langle X, \theta_0\rangle = \frac{ay_0(X)}{t_0(X)}$ and $\mathbb{E}[Y|X] = y_0(X)$.

Thus, we can show that $D_g m(\theta_0, g_0, X, Y, T)[\hat{g} - g_0] = 0$

The continuity of the second order functional derivative naturally follows. $\square$

The following assumption comes from the assumption in Theorem B.14.

**Assumption B.16** (Invertibility of the Jacobian). We assume that the Jacobian matrix satisfies

$$\|J(t_o)^{-1}\|_{\text{op}} = \Big\|\mathbb{E}\Big[-a^{-2}t_o(X)^2\, XX^T\Big]\Big\|_{\text{op}} \leq C,$$

for some constant $C > 0$. This claim is justified under mild conditions on the boundedness of the reward and eigen values of the data matrix $\mathbb{E}[XX^T]$.

Furthermore, we assert the following claim regarding the boundedness of the nuisance function $t_o(X)$.

We now state our assumptions on the convergence rates of the nuisance function estimators.

**Claim B.17** (Nuisance Estimation Rates). *There exists a black-box learner such that its root mean squared error (RMSE) satisfies*

$$\|\hat{t} - t_o\|_{L_2(\mathcal{D})} = o\Big(\frac{1}{n^{\beta_1}}\Big) \quad \text{with } \beta_1 \geq \frac{1}{4},$$

*and*

$$\|\hat{y} - y_o\|_{L_2(\mathcal{D})} = o\Big(\frac{1}{n^{\beta_2}}\Big) \quad \text{with } \beta_2 \geq \frac{1}{2} - \beta_1.$$

*These conditions ensure that the nuisance estimates converge sufficiently fast as the sample size $n$ increases.*

A discussion on how these slow rates can be attained in provided in Appendix B.5.

**Claim B.18.**

$$t_o(X) = \frac{\tanh(\langle \theta_o, X \rangle)}{\langle \theta_o, X \rangle} \leq 1.$$

A brief inspection using the properties of the hyperbolic tangent function (notably, that $\tanh(x) \leq x$ for $x \geq 0$) confirms this bound.

We now show that assumptions in Assumption B.4 hold for our loss function using the following four claims and lemmas namely Claim B.19, Claim B.20, Lemma B.21 and Claim B.22.

**Claim B.19.** *The jacobian $J(\cdot)$ satisfies $J(\hat{t}) - J(t_o) = o(1)$ and the $q^{th}$ order moment conditions hold i.e. i.e. $\sup_{g \in \mathcal{G}} \left( \mathbb{E}\left[ ||m(\theta_o, g, Z)||^q \right]^{1/q} \right)$ and $\sup_{g \in \mathcal{G}} \left( \mathbb{E}\left[ ||j(g, Z)||^q \right]^{1/q} \right)$ is bounded.*

*Proof.* Write

$$J(t) - J(t_o) = a^{-2} \mathbb{E}\left[ \left( t(X)^2 - t_o(X)^2 \right) X X^T \right].$$

Since the operator norm is bounded by the Frobenius norm, we have

$$\|J(t) - J(t_o)\|_{op} \leq \|J(t) - J(t_o)\|_F.$$

Using the Frobenius norm, we estimate

$$\|J(t) - J(t_o)\|_F = a^{-2} \left\| \mathbb{E}\left[ \left( t(X)^2 - t_o(X)^2 \right) X X^T \right] \right\|_F.$$

Using the boundedness of $X$ (i.e. $\|X\| \leq 1$), we have

$$\|X X^T\|_F = \|X\|^2 \leq 1.$$

Notice that

$$|t(X)^2 - t_o(X)^2| = |t(X) - t_o(X)| \, |t(X) + t_o(X)|.$$

Both $t$ and $t_o$ are uniformly bounded ($|t(X)|, |t_o(X)| \leq 1$), then $|t(X) + t_o(X)| \leq 2$. Hence, $|t(X)^2 - t_o(X)^2| \leq 2 |t(X) - t_o(X)|$. Thus, we obtain

$$\|J(t) - J(t_o)\|_F \leq a^{-2} \mathbb{E}\left[ 2 |t(X) - t_o(X)| \cdot \right] = 2a^{-2} \mathbb{E}\left[ |t(X) - t_o(X)| \right].$$

Finally, applying Jensen's inequality (or noting that $\mathbb{E}[|t(X) - t_o(X)|] \leq \|t - t_o\|_{L_2(\mathcal{D})}$) yields

$$\|J(t) - J(t_o)\|_{op} \leq 2a^{-2} \|t - t_o\|_{L_2(\mathcal{D})}.$$

This concludes the proof of the first part as the nuisance estimators are consistent i.e $||\hat{g} - g_o||_{L_2(\mathcal{D})} = o_p(1)$

We now check the moment condition. This naturally follows from the fact that $\mathbb{E}[T^\alpha \mid X]$ is bounded for every $\alpha \geq 1$ and the fact that rest all random variables or functions are bounded.

$\square$

**Claim B.20.** *Let $g$ be a vector-valued function defined as*

$$g(x) = \begin{bmatrix} t(x) \\ y(x) \end{bmatrix}.$$

*Then, for any $\bar{g} = \tau g + (1 - \tau)g_o$ with $\tau \in [0, 1]$, we have*

$$\sqrt{n} \, D_{gg} M(\theta_o, \bar{g})[g - g_o] = o_p(1).$$

*Proof.* We wish to control the second-order term in the expansion of $M(\theta_o, g)$ around $g_o$. By Taylor's theorem in the direction $g - g_o$, we have

$$D_{gg}M(\theta_o, \bar{g})[g - g_o] = \frac{\partial^2}{\partial s^2}M\Big(\theta_o, \bar{g} + s(g - g_o)\Big)\Big|_{s=0}.$$

This term decomposes into two parts:

$$D_{gg}M(\theta_o, \bar{g})[g - g_o] = \underbrace{\frac{\partial^2}{\partial s^2}\Big(J(\bar{t} + s(t - t_o))\theta_o\Big)\Big|_{s=0}}_{(I)} + \underbrace{\frac{\partial^2}{\partial s^2}V\Big(\bar{t} + s(t - t_o), \bar{y} + s(y - y_o)\Big)\Big|_{s=0}}_{II}.$$

(33)

First Term (I): For $J(t) = a^{-2}\mathbb{E}[t(X)^2 X X^T]$, we have

$$\frac{\partial^2}{\partial s^2}\Big((\bar{t}(X) + s(t(X) - t_o(X)))^2\Big)\Big|_{s=0} = 2(t(X) - t_o(X))^2.$$

Thus,

$$\frac{\partial^2}{\partial s^2}\Big(J(\bar{t} + s(t - t_o))\theta_o\Big)\Big|_{s=0} = 2a^{-2}\mathbb{E}\Big[(t(X) - t_o(X))^2 X X^T\Big]\theta_o.$$

Since $\|X X^T\|_{\mathrm{op}}$ is bounded (using $\|X\| \leq 1$) and $\theta_o$ is fixed, it follows that

$$\Big\|2\mathbb{E}\Big[(t(X) - t_o(X))^2 X X^T\Big]\theta_o\Big\| = O\Big(\|t - t_o\|_2^2\Big).$$

Under Claim B.17 we have $\|t - t_o\|_2^2 = o\Big(\frac{1}{\sqrt{n}}\Big)$, so that

$$\sqrt{n}\,O\Big(\|t - t_o\|_2^2\Big) = o(1).$$

Second Term (II): For the function $V(t, y)$, we have

$$\frac{\partial^2}{\partial s^2}V\Big(\bar{t} + s(t - t_o), \bar{y} + s(y - y_o)\Big)\Big|_{s=0} = a^{-1}\mathbb{E}\Big[(t(X) - t_o(X))(y(X) - y_o(X))X\Big].$$

Using the boundedness of $X$ and applying the Cauchy–Schwarz inequality, we obtain

$$\Big\|\mathbb{E}\Big[(t(X) - t_o(X))(y(X) - y_o(X))X\Big]\Big\| \leq C\,\|t - t_o\|_{L_2(\mathcal{D})}\,\|y - y_o\|_{L_2(\mathcal{D})},$$

for some constant $C > 0$. By Claim B.17, the product $\|t - t_o\|_{L_2(\mathcal{D})}\,\|y - y_o\|_{L_2(\mathcal{D})}$ is $o\Big(\frac{1}{\sqrt{n}}\Big)$. Therefore,

$$\sqrt{n}\,\Big\|\mathbb{E}\Big[(t(X) - t_o(X))(y(X) - y_o(X))X\Big]\Big\| = o(1).$$

Combining the two terms, we conclude that

$$\sqrt{n}\,D_{gg}M(\theta_o, \bar{g})[g - g_o] = o_p(1).$$

This completes the proof. $\qquad\square$

**Lemma B.21.** *The following holds*

$$\mathbb{E}[\|m(\theta_o, y, t; X, Y, T) - m(\theta_o, y_o, t_o; X, Y, T)\|^2] = O\Big(\|t - t_o\|^2{}_{L_2(\mathcal{D})} + \|y - y_o\|_{L_2(\mathcal{D})}^2\Big)$$

. *Given that* $\|\hat{t} - t_o\|_{L_2(\mathcal{D})} = o(1)$ *and* $\|\hat{\mathfrak{r}} - r_o\|_{L_2(\mathcal{D})} = o(1)$, *we have* $\mathbb{E}[\|m(\theta_o, \hat{g}, Z) - m(\theta_o, g_0; Z)\|^2] = o(1)$

*Proof.* We have

$$m(\theta_o, y, t; X, Y, T) - m(\theta_o, y_o, t_o; X, Y, T)$$
$$= a^{-1}\Big(a^{-1}\langle\theta_o, X\rangle(t(X)^2 - t_o(X)^2) + (y(X) - y_o(X))T + y_0(X)t_o(X) - y(X)t(X) + Y(t_o(X) - t(X))\Big)X$$

We can further write the term

$$y_o(X)t_o(X) - y(X)t(X) = (y_o(X) - y(X))t_o(X) + y(X)(t_o(X) - t(X))$$

Using the fact that $||X||_2 \leq 1$, we have

$$\mathbb{E}\Big(m(\theta_o, y, t; X, Y, T) - m(\theta_o, y_o, t_o; X, Y, T)\Big)_i^2 = \sqrt{d}a^{-1}\mathbb{E}\bigg[\Big(a^{-1}\langle\theta_o, X\rangle(t(X) - t_o(X))(t(X) + t_o(X))$$

$$+ (y(X) - y_o(X))(T - t_o(X)) + y(X)(t_o(X) - t(X)) + Y(t_o(X) - t(X))\Big)^2\bigg]$$

The desired result is immediate via the the identity that $(a + b + c + d)^2 \leq 4(a^2 + b^2 + c^2 + d^2)$ except for the following term

$$\mathbb{E}\left[\Big((y(X) - y_o(X))^2(T - t_o(X))\Big)^2\right] = \mathbb{E}\left[\mathbb{E}\left[\Big((y(X) - y_o(X))^2(T - t_o(X))\Big)^2 \mid X\right]\right]$$

$$= \mathbb{E}\left[(y(X) - y_o(X))^4\mathbb{E}\left[\Big((T - t_o(X))\Big)^2 \mid X\right]\right]$$

$$= \mathbb{E}\left[(y(X) - y_o(X))^2\text{Var}(T \mid X)\right] \leq \mathbb{E}\left[(y(X) - y_o(X))^2\right]$$

The last inequality follows since $\text{Var}(T \mid X) \leq 1$. Finally this gives us

$$\mathbb{E}\Big(m(\theta_o, y, t; X, Y, T) - m(\theta_o, y_o, t_o; X, Y, T)\Big)_i^2 \leq (4S + 1)||t - t_o||^2_{L_2(\mathcal{D})} + (4S + 1)||y - y_o||^2_{L_2(\mathcal{D})}$$

$\square$

**Claim B.22.** *Each component of the matrix $j(t_o, X)$ satisfies*

$$\mathbb{E}\Big[j_{pq}(t_o, X)^2\Big] = O(1).$$

*Furthermore, the components of the moment function satisfy*

$$\mathbb{E}\Big[m_q(\theta_o, y_o, t_o)^2\Big] = O(1).$$

*Proof.* The first statement follows directly from the boundedness of $t_o(X)$ and the boundedness of $X$. For the second statement, note that

$$\mathbb{E}\Big[m_q(\theta_o, y_o, t_o)^2\Big] \leq 2a^{-2}\left(\mathbb{E}\Big[t_o(X)^2\Big] + \mathbb{E}\Big[a^{-2}\langle X, \theta_o\rangle^2 t_o(X)^2\Big] \cdot \mathbb{E}\Big[y_o(X)\big(T - t_o(X)\big)^2\Big]\right).$$

Since the functions $y_o$ and $t_o$ are bounded and the variance of $T$ is also bounded , it follows that

$$\mathbb{E}\Big[m_q(\theta_o, y_o, t_o)^2\Big] = O(1).$$

$\square$

Invoking Lemma B.5 obtained via minimising orthogonal loss $\mathcal{L}^{\text{ortho}-2}$ is asymptotically linear. In particular,

$$\sqrt{n}\left(\hat{\theta}_{\text{ortho}} - \theta_o\right) = -\frac{1}{\sqrt{n}}\sum_{i=1}^n IF(\theta_o, y_o, t_o; X_i, Y_i, T_i) + o_p(1), \tag{34}$$

where the influence function is given by

$$IF(\theta_o, y_o, t_o; X_i, Y_i, T_i) = \mathbb{E}[a^{-2}t_o(X)^2 XX^T]^{-1}m(\theta_o, y_o, t_o; X_i, Y_i, T_i)$$

Note that by the definition of $\theta_o$ we have $\mathbb{E}\left[m(\theta_o, y_o, t_o; X, Y, T)\right] = 0$. Now

$$m(\theta_o, t_o, y_o; X, Y, T) = \frac{1}{a}\Big(\frac{\langle X, \theta\rangle}{a}t_o(X)^2 + y_o(X)\big(T - t_o(X)\big) - Y t_o(X)\Big)X \tag{35}$$

$$= \frac{1}{a}\big(y_o(X)\big(T - t_o(X)\big) - t_o(X)\big(Y - y_o(X)\big)\big) \tag{36}$$

Thus using the fact that $\mathbb{E}[T \mid X] = t_o(X), \mathbb{E}[Y \mid X] = y_o(X)$ and $\text{Var}(Y \mid X) = 1 - y_o(X)^2$, we get

$$\text{Covar}(m(\theta_o, y_o, t_o; X, Y, T)) = \frac{1}{a^2}\mathbb{E}\left[\left(t_o(X)^2(1 - y_o(X)^2) + \text{Var}(T|X) \cdot y_o(X)^2\right)XX^T\right] \tag{37}$$

$$= \frac{1}{a^4}\mathbb{E}\left[t_o(X)^3 XX^T\right] \tag{38}$$

The last step follows from the fact that (Claim B.24). Thus, the covariance of the influence function is given by

$$\text{Covar}(IF) = \mathbb{E}[t_o(X)^2 XX^T]^{-2}\mathbb{E}[t_o(X)^3 XX^\top]$$

This completes the proof of theorem B.14.

### B.5 Estimation of nuisance functions in first stage

**Plug-in loss**:

In this setup, one can first estimate $\mathfrak{r}(\cdot)$ by minimizing the log-loss as described in Appendix B and thus obtain $||\hat{\mathfrak{r}}(\cdot) - r(\cdot)||_{L_2(\mathcal{D})} = O\left(\frac{e^S}{\sqrt{n}}\right)$. Now plug in this to estimate $\hat{t}(\cdot) = \frac{\tanh \mathfrak{r}(\cdot)}{\mathfrak{r}(\cdot)}$, we can get $||\hat{t} - t_o||_{L_2(\mathcal{D})} = O\left(\frac{e^S}{\sqrt{n}}\right)$ using Lipschitzness.

For orthogonal loss $\mathcal{L}^{\text{ortho}-2}$, a similar plug-in would give $||\hat{y} - y_o||_{L_2(\mathcal{D})} = O\left(\frac{e^S}{\sqrt{n}}\right)$

This provides the slow rates that we desire in Claim B.8 and Claim B.17 for orthogonal losses $\mathcal{L}^{\text{ortho}}$ and $\mathcal{L}^{\text{ortho}-2}$ respectively.

**Separate nuisance estimation**:

In this setup, we separately estimate the nuisance function $t(\cdot)$ and $\mathfrak{r}(\cdot)$. The reward nuisance function $\mathfrak{r}(\cdot)$ can be estimated by minimizing the log-loss as described above to obtain $||\hat{\mathfrak{r}}(\cdot) - r(\cdot)||_{L_2(\mathcal{D})} = O\left(\frac{e^S}{\sqrt{n}}\right)$.

To estimate time, we get first argue that the $W_{s,2}$ Sobolev norm [AF03] (with $s \geq 1$) of $t_o$ is bounded using the composition theorem [Mos66]. More specifically, one can show that $W_{s,2}$ Sobolev norm of $t_o(\cdot)$ is given by $O\left(Ss!\right)$ as the $s^{th}$ order derivative of $\frac{\tanh x}{x}$ scales as $s!$. Now applying kernel ridge regression with a Sobolev kernel associated with the RKHS $W^{s,2}$ [AF03, Wai19] to get the desired slow rate i.e. $||\hat{t} - t_o||_{L_2(\mathcal{D})} = O\left(s!Sn^{-\frac{2s}{2s+d}}\right)$. One can choose the value of $s$ appropriately to obtain the desired rates. Choosing $s$ sufficiently high can give us faster decay with number of samples $n$ but the pre-muliptlier $s!$ would be higher.

For $\mathcal{L}^{\text{ortho}-2}$, one can bound $||\hat{y} - y_o|| = O\left(\frac{e^S}{\sqrt{n}}\right)$ via plug-in.

We thus get the desired slow rates to satisfy Claim B.8 and Claim B.17

### B.6 Inequalities used in asymptotic results

We now prove two inequalities that have been used in the main paper. However, we restrict ourselves to the case where $a = 1$ and $t_{\text{nondec}} = 0$.

**Claim B.23.** *Var*$(T \mid X) \leq (\mathbb{E}[T \mid X])^2$

*Proof.* Now consider the expression of $\text{Var}(T \mid X)$ from Appendix A.1. Now denoting $r(X)$ by $r$, we have

$$\text{Var}(T \mid X) = a\frac{e^{4ar} - 1 - 4are^{2ar}}{r^3(e^{2ar} + 1)^2} \text{ and } (\mathbb{E}[T \mid X])^2 = \frac{a^2(e^{2ar} - 1)^2}{r^2(e^{2ar} + 1)^2} \tag{39}$$

Observe that it is sufficient to consider $r \geq 0$ as both functions are symmetric. Thus, it is sufficient to show that $ar(e^{2ar} - 1)^2 \geq e^{4ar} - 1 - 4are^{2ar} \ \forall r \geq 0$. Now define $v = ra$ and we now define the function $f(v)$ as follows and argue that it is non-negative.

$$f(v) := v(e^{2v} - 1)^2 - e^{4v} - 1 - 4ve^{2v} = (v - 1)e^{4v} + 2ve^{2v} + v + 1 \tag{40}$$
$$= \left(v(e^{2v} + 1) - (e^{2v} - 1)\right)(e^{2v} + 1) \tag{41}$$

Since $e^{2v} + 1 > 0$, it suffices to show

$$g(v) := v(e^{2v} + 1) - (e^{2v} - 1) \geq 0.$$

Differentiating w.r.t. $v$ gives

$$g'(v) = (2v - 1)e^{2v} + 1, \qquad g''(v) = 4ve^{2v} \geq 0.$$

Hence $g'(v)$ is monotonically increasing, so

$$g'(v) \geq g'(0) = 0,$$

which in turn implies

$$g(v) \geq g(0) = 0.$$

Thus the desired result follows.

$\square$

**Claim B.24.** *Define $y_o(X) = \mathbb{E}[T \mid X]$ and $t_o(X) = \mathbb{E}[T \mid X]$. We then have*

$$t_o(X)^2(1 - y_o(X)^2) + \text{Var}(T \mid X)y_o(X)^2 = \frac{t_o(X)^3}{a^2}$$

*Proof.* This follows from the expressions in Appendix A.1. For brevity, we refer to $r(X)$ by $r$ in this proof.

Now consider

$$t_o(X)^2(1 - y_o(X)^2) + \text{Var}(T \mid X)y_o(X)^2 \tag{42}$$

$$= \frac{a^2}{r^2}\left(\frac{\exp(2ar) - 1}{\exp(2ar) + 1}\right)^2\left(\frac{4\exp(2ar)}{(\exp(2ar) + 1)^2}\right) + \frac{a}{r^3}\left(\frac{\exp(4ar) - 1 - 4ar\exp(2ar)}{(\exp(2ar) + 1)^2}\right)\left(\frac{\exp(2ar) - 1}{\exp(2ar) + 1}\right)^2 \tag{43}$$

$$= \frac{a}{r^3}\frac{(\exp(4ar) - 1)(\exp(2ar) - 1)^2}{(\exp(2ar) + 1)^4} = \frac{t_o(X)^3}{a^2} \tag{44}$$

$\square$

The following claim would be useful to lower bound $4a^2\sigma(2ar_o(X))\sigma(-2r_o(X))$ by $\frac{t_o(X)^2}{a^2}$.

**Claim B.25.** $4a^2\sigma(2ax)\sigma(-2ax) \leq \left(\frac{\tanh(ax)}{ax}\right)^2$ *for every $x, a \geq 0$ where $\sigma(.)$ denotes the sigmoid function.*

*Proof.* Since both functions are symmetric, it is sufficient to consider the case where $t \geq 0$. Observe that

$$\sigma(2ax)\sigma(-2ax) = \frac{e^{2ax}}{(e^{2ax} + 1)^2} \quad \text{and} \quad \left(\frac{\tanh(ax)}{ax}\right)^2 = \left(\frac{e^{2ax} - 1}{a^2x^2(e^{2ax} + 1)}\right)^2 \tag{45}$$

Now substitute $y = ax$ and thus, to prove the desired result, it is sufficient to show that $e^{2y} - 1 \geq 2ye^y \Leftrightarrow e^y - e^{-y} - 2y \geq 0$. Now define $g(y) = e^y - e^{-y} - 2y$ and thus, $g'(y) = e^y + e^{-y} - 2 \geq 0$ from AM-GM inequality. This implies $g(y) \geq g(0) = 0$ proving the desired result.

$\square$

## C  Finite sample rates for General reward functions

Given a set of $n$ samples $\{x_i\}_{i=1}^n$, each drawn i.i.d from $\mathcal{D}$, consider the empirical distribution $\mathbb{P}_n(x) := \frac{1}{n}\sum_{i=1}^n \delta_{x_i}(x)$ which places point mass $\frac{1}{n}$ at each sample. Thus, the emperical mean is denoted by $\mathbb{P}_n(f) = \frac{1}{n}\sum_{i=1}^n f(x_i)$ and the population mean $\mathbb{P}f = \mathbb{E}_{X\sim\mathcal{D}}[f(X)]$.

As noted in Section 5, our critical-radius argument for bounding the excess risk $\epsilon(\hat{r},\hat{g})$ requires the per-sample loss to be uniformly bounded. In the EZ-diffusion model, however, the response time $T$ is in principle unbounded—despite having finite moments [WY08]—so we replace the original loss by a truncated version. This truncation both restores the boundedness needed for the theory and reflects the fact that, in practice, decision times are effectively bounded.

By Markov's inequality and the boundedness of higher moments, for any $\zeta \geq 1$ we have

$$\mathbb{P}(T > \breve{B} \mid X) \leq \frac{\mathbb{E}[T^\zeta]}{\breve{B}^\zeta} \leq \frac{M(\zeta)}{\breve{B}^\zeta} . \tag{46}$$

Here $M(\zeta) = \mathbb{E}[T^\zeta]$ denotes the $\zeta$-th moment of the decision time $T$ [WY08]. One selects $\breve{B}$ to meet a desired error tolerance, ensuring the tail probability in (46) is sufficiently small. Finally, define the truncated response time $\breve{T} = \min\{T, \breve{B}\}$ and substitute $\breve{T}$ for $T$ in the loss function $\mathcal{L}^{\texttt{ortho}}$.

### C.1  Bounding the loss function $\mathcal{L}^{\texttt{ortho}}$ with bounded decision time

We now introduce some new notations with respect to the decision time. Let $\breve{B}$ denote the bound at which the decision time is capped. Let $\breve{t}_o(X) = \mathbb{E}[\breve{T} \mid X]$. Further, let $\breve{t}$ denote the nuisance function corresponding the the capped decision time and let $\overset{\circ}{\breve{t}}$ denote an estimate of $\breve{t}_o$. We further overwrite $Z$ to denote the tuple of random variables $(X, Y, \breve{T})$. The joint nuisance pair is denoted by $g = (\mathfrak{r}, \breve{t})$ and let $g_o = (\mathfrak{r}, \breve{t}_o)$. Now, assume that $\breve{t}_o \in \mathcal{T}$ and $\mathcal{G}$ denotes the joint nuisance class $\mathcal{R} \times \mathcal{T}$ with $g \in \mathcal{G}$. Before, we start the analysis we state the following bound on $t_o - \breve{t}_o$. To bound, we use the following result i.e. $\mathbb{E}[X] = \int_x \mathbb{P}(X \geq x)dx$ for a non-negative random variable $X$.

Recall that $S$ denotes an absolute bound on the reward function $r(\cdot)$ which we assume to be larger than 4.

$$\|t_o - \breve{t}_o\|_{L_\infty} = \int_{z=\breve{B}}^\infty \mathbb{P}(T > z)dz \leq \frac{M(\zeta)}{\zeta \breve{B}^{\zeta-1}} \tag{47}$$

The first equality follows as $T - \breve{T} > z$ implies $T \geq \breve{B} + z$ for every positive $z$.

Thus the orthogonalized loss function from (6) is redefined as

$$\breve{\mathcal{L}}^{\texttt{ortho}}(r, \mathfrak{r}, \breve{t}) = \mathbb{E}\left[\left(Y - (\breve{T} - \breve{t}(X))\mathfrak{r}(X) - r(X)\breve{t}(X)\right)^2\right] \tag{48}$$

Next, we verify that each of the four assumptions in [FS23, Section 3] holds for the new loss $\breve{\mathcal{L}}^{\texttt{ortho}}$. Observe that the first assumption of neyman-orthogonality satisfied approximately here with a bias decaying with the threshold $\breve{B}$ that is characterized by $\|t_o(\cdot) - \breve{t}_o(\cdot)\|_{L_\infty}$ (Equation (47)).

**Lemma C.1** (Approximately Orthogonal loss). *For all $r \in \mathcal{R}$ and $g \in \mathcal{G}$, we have*

$$|D_g D_r \breve{\mathcal{L}}^{\texttt{ortho}}(r_o, g_o)[r - r_o, g - g_o]| \leq S^2 \|t_o - \breve{t}_o\|_{L_\infty} \|\breve{t} - \breve{t}_o\|_{L_2(\mathcal{D})} \|r - r_o\|_{L_2(\mathcal{D})} \tag{49}$$

*Proof.* Consider

$$\left|D_g D_r \breve{\mathcal{L}}^{\texttt{ortho}}(r_o, g_o)[r - r_o, g - g_o]\right|$$

$$\overset{(a)}{=} \left|2\mathbb{E}\left[(\breve{T} - \breve{t}_o(X))\breve{t}_o(X)(r(X) - r_o(X))(\mathfrak{r}(X) - r_o(X))\right] - 2\mathbb{E}\left[(Y - \breve{t}_o(X)r_o(X))(\breve{t}(X) - \breve{t}_o(X))(r(X) - r_o(X))\right]\right|$$

$$\leq S^2 \mathbb{E}\left[|(t_o(X) - \breve{t}_o(X))(\breve{t}(X) - \breve{t}_o(X))(r(X) - r_o(X))|\right]$$

$$\leq S^2 \|t_o - \breve{t}_o\|_{L_\infty} \|\breve{t} - \breve{t}_o\|_{L_2(\mathcal{D})} \|r - r_o\|_{L_2(\mathcal{D})}$$

Observe that the first term in $(a)$ goes to zero via conditioning on $X$ as $\mathbb{E}[\check{T} \mid X] = \check{t}_o(X)$. The second term in $(a)$ is simplified using Cauchy-Schwarz.

□

**Lemma C.2** (First order condition). $\left|D_r\check{\mathcal{L}}^{\mathrm{ortho}}(r_o, g_o)[r - r_o]\right| \leq 2S||t_o - \check{t}_o||_{L_2(\mathcal{D})}||r - r_o||_{L_2(\mathcal{D})}$

*Proof.* Now consider the following functional derivative

$$\left|D_r\check{\mathcal{L}}^{\mathrm{ortho}}(r_o, g_o)[r - r_o]\right| = \left|2\mathbb{E}\left[(Y - (\check{T} - \check{t}_o(X))r_o(X) - r_o(X)\check{t}_o(X))\check{t}_o(X)(r(X) - r_o(X))\right]\right|$$

$$\overset{(a)}{\leq} 2S\mathbb{E}\left[\left|(t_o(X) - \check{t}_o(X)))(r(X) - r_o(X))\right|\right]$$

$$\leq 2S||t_o - \check{t}_o||_{L_2(\mathcal{D})}||r - r_o||_{L_2(\mathcal{D})}$$

$(a)$ follows via conditioning on $X$ and the fact that $\mathbb{E}[Y|X] = t_o(X)r_o(X)$. .

□

We now prove the smoothness condition from [FS23, Assumption 3]. In this setup, $||.||_{\mathcal{G}}$ is defined from (9) (denoted by $(\mathcal{R}, \alpha)$).

**Lemma C.3** (Higher-Order Smoothness of $\check{\mathcal{L}}^{\mathrm{ortho}}$). *Let $\beta_1 = 2$ and $\beta_2 = 4S$. Then $\check{\mathcal{L}}^{\mathrm{ortho}}$ satisfies:*

1.  Second-order smoothness w.r.t. target. *For all $r \in \mathcal{R}$ and all $\bar{r} \in \mathrm{star}(\mathcal{R}, r_o)$,*

$$D_r^2\check{\mathcal{L}}^{\mathrm{ortho}}(\bar{r}, g_0)[r - r_o, r - r_o] \leq \beta_1 \|r - r_o\|_{L_2(\mathcal{D})}^2.$$

2.  Higher-order smoothness. *There exists $c \in [0, 1]$ such that for all $r \in \mathrm{star}(\mathcal{R}, r_o)$, $g \in \mathcal{G}$, and $\bar{g} \in \mathrm{star}(\mathcal{G}, g_0)$,*

$$\left|D_g^2 D_r\check{\mathcal{L}}^{\mathrm{ortho}}(r_o, \bar{g})[r - r_o, g - g_0, g - g_0]\right| \leq \beta_2 \|r - r_o\|_{L_2(\mathcal{D})}^{1-r} \|g - g_0\|_{\mathcal{G}}^2.$$

*Proof.* In this proof, $\ell(\cdot)$ denotes the pointwise version of the orthogonal loss $\check{\mathcal{L}}^{\mathrm{ortho}}$.

First, observe that $D_r^2\ell(\bar{r}, g_o, X, Y, T)[r - r_o, r - r_o] = 2t_o^2(X)(r(X) - r_o(X))^2$. Thus, $\mathbb{E}\left[D_r^2\ell(\bar{r}, g_o, X, Y, T)[r - r_o, r - r_o]\right] \leq 2||\theta - \theta_o||_2^2$ where $r_o(x) = \langle\theta_o, x\rangle$ for every $x \in \mathbb{R}^{2d}$. The last inequality follows from the fact that $t_o(X_1, X_2) \leq 1$.

$$\mathbb{E}\left[D_g^2 D_r\ell(r_o, \bar{g}, X, Y, T)[r - r_o, g - g_o, g - g_o]\right]$$
$$= \mathbb{E}\left[2r_o(X)(t(X) - t_o(X))^2(r(X) - r_o(X)) - 4t_o(X)(t(X) - t_o(X))(\mathfrak{r}(X) - r_o(X))(r(X) - r_o(X))\right]$$
$$\tag{50}$$
$$\leq 4S||g - g_o||_{(\mathcal{R}, \alpha)}^2||r - r_o||_{L_2(\mathcal{D})}^{1-\alpha} \tag{51}$$
$$\tag{52}$$

The last statement follows from the definition.

□

**Lemma C.4** (Strong Convexity of $\check{\mathcal{L}}^{\mathrm{ortho}}$). *The population risk is strongly convex w.r.t. the target parameter. There exist constant $0 < \lambda \leq \left(\frac{\tanh(S)}{S}\right)^2$ such that for all $r \in \mathcal{R}$, $g \in \mathcal{G}$ and all $\bar{r} \in \mathrm{star}(\mathcal{R}, r_o)$,*

$$D_r^2 L_{\mathcal{D}}(\bar{r}, g)[r - r_o, r - r_o] \geq \lambda \|r - r_o\|_{L_2(\mathcal{D})}^2$$

*where $r \in [0, 1]$ is as in Assumption C.3.*

*Proof.* First observe that $D_r^2 \ell(\bar{r}, g, X, Y, T)[r - r_o, r - r_o] = 2t^2(X)(r(X) - r_o(X))^2$. Thus,

$$\mathbb{E}\left[D_r^2 \ell(\bar{r}, g, X, Y, T)[r - r_o, r - r_o]\right] = 2\mathbb{E}[t^2(X)(r(X) - r_o(X))^2] \tag{53}$$

Now considering a lower bound of $\frac{\tanh(S)}{S}$ on every function in $t \in \mathcal{T}$, we prove the lemma. $\qquad\square$

With this, we now prove the main theorem below using [FS23, Theorem 1].

**Theorem C.5.** *Suppose $\hat{r}$ minimizes the orthogonal loss $\check{\mathcal{L}}^{\text{ortho}}$ and satisfies*

$$\check{\mathcal{L}}^{\text{ortho}}(\hat{r}, \hat{g}) - \check{\mathcal{L}}^{\text{ortho}}(r_o, \hat{g}) \leq \epsilon(\hat{r}, \hat{g}).$$

*With $S$ denoting an absolute bound on $r_o$. Then the two stage meta-algorithm 1 with orthogonal loss $\check{\mathcal{L}}^{\text{ortho}}$ guarantees*

$$||\hat{r} - r_o||^2_{L_2(\mathcal{D})} \leq CS^2\left(\epsilon(\hat{r}, \hat{g}) + ||t_o - \check{t}_o||^2_{L_\infty}\right) + 4S^{\frac{4}{1+\alpha}}||\hat{g} - g_o||^{\frac{4}{1+\alpha}}_{(\mathcal{R}, \alpha)} \tag{54}$$

*Further, the error term $||t_o - \check{t}_o||_{L_\infty}$ can be bounded in (47) in terms of bound $\check{B}$. One can choose moment $\zeta \geq 1$ to get the largest bound.*

*Proof.* Applying [FS23] [6], we observe that

$$||\hat{r} - r_o||^2_{L_2(\mathcal{D})}$$
$$\leq 4S^2\left(\epsilon(\hat{r}, \hat{g}) + 2S||t_o - \check{t}_o||_{L_2(\mathcal{D})}||\hat{r} - r_o||_{L_2(\mathcal{D})} + S^2||t_o - \check{t}_o||_{L_\infty}||\check{t} - \check{t}_o||_{L_2(\mathcal{D})}||\hat{r} - r_o||_{L_2(\mathcal{D})}\right)$$
$$+ 4S^{\frac{4}{1+\alpha}}||\hat{g} - g_o||^{\frac{4}{1+\alpha}}_{(\mathcal{R}, \alpha)} \tag{55}$$

Applying the AM-GM inequality, we have for some constant $C$

$$||\hat{r} - r_o||^2_{L_2(\mathcal{D})} \leq CS^2\epsilon(\hat{r}, \hat{g}) + S^2||t_o - \check{t}_o||^2_{L_\infty}(1 + |\check{t} - \check{t}_o||^2_{L_2(\mathcal{D})}) + 4S^{\frac{4}{1+\alpha}}||\hat{g} - g_o||^{\frac{4}{1+\alpha}}_{(\mathcal{R}, \alpha)} \tag{56}$$

$$\leq CS^2\left(\epsilon(\hat{r}, \hat{g}) + ||t_o - \check{t}_o||^2_{L_\infty}\right) + 4S^{\frac{4}{1+\alpha}}||\hat{g} - g_o||^{\frac{4}{1+\alpha}}_{(\mathcal{R}, \alpha)} \tag{57}$$

$\qquad\square$

We now prove the following corollaries from Section 5. As we discussed above, we cannot instantiate Theorem 5.1 directly to obtain error rates using critical radius as the critical radius crucially invokes Talagrand's inequality which requires the loss functions to be point-wise bounded. We thus invoke Theorem C.5 with bounded loss function $\check{\mathcal{L}}^{\text{ortho}}$ by capping the decision time $T$ by a threshold $\check{B}$.

## C.2  Proof of Corollary 5.3 and Corollary 5.4

**Corollary 5.3** (Data-splitting). *Let $\delta_n$ be the critical radius of the star-shaped class*

$$\text{star}\{r - r_o : r \in \mathcal{R}, 0\},$$

*and define $\delta_n^{\text{ds}} = \max\{\delta_n, \sqrt{c/n}\}$ for some constant $c > 0$. Then under data-splitting, with probability at least $1 - c_1 \exp(-c_2 n(\delta_n^{\text{ds}})^2)$, Meta-Algorithm 1 satisfies*

$$\epsilon(\hat{r}, \hat{g}) \leq 9\,S\,\check{B}\delta_n^{\text{ds}}\,||\hat{r} - r_o||_{L_2(\mathcal{D})} + 10\,S^2\check{B}\,(\delta_n^{\text{ds}})^2,$$

*for universal constants $c_1, c_2 > 0$. Consequently for every $\zeta \geq 1$,*

$$||\hat{r} - r_o||^2_{L_2(\mathcal{D})} \leq \text{poly}(S)\left(\check{B}\,(\delta_n^{\text{ds}})^2 + \left(\frac{M(\zeta)}{\zeta\check{B}^{\zeta-1}}\right)^2\right) + 4\,S^{\frac{4}{1+\alpha}}\,||\hat{g} - g_o||^{\frac{4}{1+\alpha}}_{(\mathcal{R}, \alpha)},$$

*holds with the same probability.*

---

[6]Although [FS23] assumes exact orthogonality of the loss, any remaining bias contributes only an additive term—which we account for here. This immediately follows from the proof of [FS23, Theorem 1].

*Proof.* Let $\check{\mathcal{L}}^{\texttt{ortho}}(\cdot)$ denote the population loss and let $\check{\mathcal{L}}^{\texttt{ortho}}_{\mathcal{S}}(\cdot)$ be the empirical loss evaluated on the sample $\mathcal{S} = \{Z_1, \ldots, Z_n\}$. Further, write $\check{\ell}^{\texttt{ortho}}$ for the pointwise version of the loss $\check{\mathcal{L}}^{\texttt{ortho}}$.

It suffices to show that for every $r \in \mathcal{R}$,

$$\left| \check{\mathcal{L}}^{\texttt{ortho}}_{\mathcal{S}}(r, \hat{g}) - \check{\mathcal{L}}^{\texttt{ortho}}_{\mathcal{S}}(r_o, \hat{g}) - \left( \check{\mathcal{L}}^{\texttt{ortho}}(r, \hat{g}) - \check{\mathcal{L}}^{\texttt{ortho}}(r_o, \hat{g}) \right) \right|$$
$$\leq 9\, S\, \check{B}\, \delta^{\texttt{ds}}_n \|r - r_o\|_{L_2(\mathcal{D})} + 10\, \check{B}\, (\delta^{\texttt{ds}}_n)^2. \tag{58}$$

Once (58) holds, the fact that $\hat{r}$ minimizes the empirical loss (so $\check{\mathcal{L}}^{\texttt{ortho}}_{\mathcal{S}}(\hat{r}, \hat{g}) - \check{\mathcal{L}}^{\texttt{ortho}}_{\mathcal{S}}(r_o, \hat{g}) \leq 0$) immediately yields the desired corollary.

The proof of this analysis follows standard techniques [Wai19, Theorem 14.20]. Let $\mathscr{R}_o := \text{star}(\{r - r_o : r \in \mathcal{R}\}, 0)$ be the star-convex hull. Further, let

$$Z_n(\zeta) := \sup_{||r-r_o||_{L_2(\mathcal{D})} \leq \zeta} ||\mathbb{P}_n(\check{\ell}^{\texttt{ortho}}(r, \hat{g}, \cdot) - \check{\ell}^{\texttt{ortho}}(r_o, \hat{g}, \cdot)) - \mathbb{P}(\check{\ell}^{\texttt{ortho}}(r, \hat{g}, \cdot) - \check{\ell}^{\texttt{ortho}}(r_o, \hat{g}, \cdot))||.$$
$$\tag{59}$$

We now define two events as function of nuisance $g(.)$. Let

$$\mathcal{E}_o = \left\{ Z_n(\delta^{\texttt{ds}}_n) \geq 10 \check{B} \left( \delta^{\texttt{ds}}_n \right)^2 \right\}$$

and

$$\mathcal{E}_1 = \left\{ \exists\, r \in \mathcal{R} \;\middle|\; \mathbb{P}_n\left( \check{\ell}^{\texttt{ortho}}(r, \hat{g}, \cdot) - \check{\ell}^{\texttt{ortho}}(r_o, \hat{g}, \cdot) \right) - \mathbb{P}\left( \check{\ell}^{\texttt{ortho}}(r, \hat{g}, \cdot) - \check{\ell}^{\texttt{ortho}}(r_o, \hat{g}, \cdot) \right) \right.$$
$$\left. \geq 9\, S\, \check{B}\, \delta^{\texttt{ds}}_n ||r - r_o||_{L_2(\mathcal{D})} \text{ and } ||r - r_o||_{L_2(\mathcal{D})} \geq \delta^{\texttt{ds}}_n \right\}.$$
$$\tag{60}$$

If the bound in (58) does not hold, then either event $\mathcal{E}_0$ or event $\mathcal{E}_1$ must be true. The probability of the event $\mathcal{E}_1$ can be bounded using same peeling arguments from [Wai19, Theorem 14.1].

To bound the probability of $\mathcal{E}_o$, we first employ Talagrand's concentration for empirical processes to obtain for every nuisance estimate $\hat{g}$ -

$$\mathbb{P}\left( Z_n(\zeta) \geq 2E[Z_n(\zeta)] + u \right) \leq c_1 \exp\left( -\frac{c_2 n u^2}{\zeta^2 + u} \right) \tag{61}$$

We shall now crucially use the fact that nuisance function $\hat{g}(.)$ is conditionally independent of the data-set $Z_1, \ldots Z_n$.

$$\mathbb{E}\left[ Z_n(\zeta) \right] \overset{(i)}{\leq} 2\mathbb{E}\left[ \sup_{r \in \mathcal{R}: ||r-r_o||_{L_2(\mathcal{D})} \leq \zeta} \frac{1}{n} \left| \sum_i \epsilon_i \left( \ell^{\texttt{ortho}}(r, \hat{g}, Z_i) - \ell^{\texttt{ortho}}(r_o, \hat{g}, Z_i) \right) \right| \right] \tag{62}$$

$$\overset{(ii)}{\leq} 2\mathbb{E}\left[ \sup_{r \in \mathcal{R}: ||r-r_o||_{L_2(\mathcal{D})} \leq \zeta} \frac{1}{n} \left| \sum_i S \check{B} \epsilon_i (r - r_o) \right| \right] \tag{63}$$

$$\overset{(iii)}{\leq} C S \check{B} \text{Rad}_n(\mathscr{R}_o, \zeta) \leq 4 S \check{B} r \delta^{\texttt{ds}}_n \text{ valid for all } \zeta \geq \delta^{\texttt{ds}}_n \tag{64}$$

The step (i) follows from symmetrization argument, step (ii) follows from the independence of $\hat{g}$ with respect to data-set $Z_1, \ldots, Z_n$ and bound $\check{B}$ on decision time $\check{T}$ and step (iii) follows from our choice of $\delta^{\texttt{ds}}_n$.

Now apply the Talagrand's concentration lemma to bound the probability of event $\mathcal{E}_o$ and conclude the proof.

The corollary statement follows by applying the bound $\epsilon(\hat{r}, \hat{g})$ to Theorem C.5 and bounding the rate $||t_o - \check{t}_o||_{L_\infty}$ using Equation (47).

$\square$

Next, we prove the corollary for data-reuse for $\check{\mathcal{L}}^{\text{ortho}}$. Observe that the critical radius is computed over a bigger class for the case of data-reuse as the independence of nuisance estimate $\hat{g}$ and the data-set $Z_1, \ldots, Z_n$ cannot be assumed.

**Corollary 5.4** (Data-reuse). *Let $\delta_n$ be the critical radius of*

$$\text{star}\{\, \ell^{\text{ortho}}(r, g; \cdot) - \ell^{\text{ortho}}(r_o, g; \cdot) : r \in \mathcal{R}, \ g \in \mathcal{G}\},$$

*and define $\delta_n^{\text{dr}} = \max\{\delta_n, \sqrt{c/n}\}$ for some constant $c > 0$. Then under data-reuse, with probability at least $1 - c_1 \exp\left(-c_2 n (\delta_n^{\text{dr}})^2\right)$, Meta-Algorithm 1 satisfies*

$$\epsilon(\hat{r}, \hat{g}) \ \leq \ 9\, S\, \delta_n^{\text{dr}} \, \|\hat{r} - r_o\|_{L_2(\mathcal{D})} \ + \ 10\, (\delta_n^{\text{dr}})^2,$$

*for universal constants $c_1, c_2 > 0$. Consequently for every $\zeta \geq 1$,*

$$\|\hat{r} - r_o\|_{L_2(\mathcal{D})}^2 \ \leq \ \text{poly}(S) \left((\delta_n^{\text{dr}})^2 + \left(\frac{M(\zeta)}{\zeta \check{B}^{\zeta - 1}}\right)^2\right) + \ 4\, S^{\frac{4}{1+\alpha}} \, \|\hat{g} - g_o\|_{(\mathcal{R}, \alpha)}^{\frac{4}{1+\alpha}},$$

*holds with the same probability.*

*Proof.* We denote $\check{\mathcal{L}}_{\mathcal{S}}^{\text{ortho}}(.)$ as the sample loss evaluated on a set of $n$ data points with $\mathcal{S} = \{Z_1, \ldots, Z_n\}$. Further, denote $\check{\ell}^{\text{ortho}}$ as the point-wise loss version of $\check{\mathcal{L}}^{\text{ortho}}$.

Observe that it is sufficient to show that the following bound is satisfied. This proves the corollary as minimizing the empirical loss ensures $\check{\mathcal{L}}_{\mathcal{S}}^{\text{ortho}}(\hat{r}, g) - \check{\mathcal{L}}_{\mathcal{S}}^{\text{ortho}}(r_o, g)) \geq 0$ for some $g \in \mathcal{G}$.

$$\left|(\check{\mathcal{L}}_{\mathcal{S}}^{\text{ortho}}(r, g) - \mathcal{L}_{\mathcal{S}}^{\text{ortho}}(r_o, g)) - \check{\mathcal{L}}^{\text{ortho}}(r, g) - \check{\mathcal{L}}^{\text{ortho}}(r_o, g))\right|$$

$$\leq 9\delta_n^{\text{dr}}|\check{\mathcal{L}}^{\text{ortho}}(r; g) - \check{\mathcal{L}}^{\text{ortho}}(r_o; g)| + 10(\delta_n^{\text{dr}})^2 \ \forall r \in \mathcal{R} \text{ and } g \in \mathcal{G}. \tag{65}$$

The proof of this analysis follows standard techniques [Wai19, Theorem 14.20]. Let $\mathscr{F}$ denote the star convex hull defined in the corollary:

$$\mathscr{F} := \text{star}\left(\{Z \to \check{\ell}^{\text{ortho}}(r; g, Z) - \check{\ell}^{\text{ortho}}(r_o; g, Z) : \ \forall r \in \mathcal{R}, g \in \mathcal{G}\}, 0\right) \tag{66}$$

In the notation below, $f$ denotes any element in $\mathcal{F}$.

$$Z_n(r) := \sup_{\|f\|_2 \leq r} \|\mathbb{P}_n(f) - \mathbb{P}(f)\|$$

Now we define two events

$$\mathcal{E}_0 = \{Z_n(\delta_n^{\text{dr}}) \geq 9\, (\delta_n^{\text{dr}})^2\}$$

and

$$\mathcal{E}_1 = \{\exists f \in \mathscr{F} \mid \mathbb{P}_n(f) - \mathbb{P}(f) \geq 10\delta_n^{\text{dr}}\|f\|_{L_2(\mathcal{D})} \text{ and } \|f\|_{L_2(\mathcal{D})} \geq \delta_n^{\text{dr}}\}$$

If that the bound in (65) does not hold true, either event $\mathcal{E}_0$ or event $\mathcal{E}_1$ must hold true. The probability of $\mathcal{E}_1$ can be bounded by same peeling arguments as done in [Wai19, Theorem 14.1].

To bound the probability of $\mathcal{E}_0$, we first employ Talagrand's concentration for empirical processes to obtain

$$\mathbb{P}\left(Z_n(\zeta) \geq 2E[Z_n(\zeta)] + u\right) \leq c_1 \exp\left(-\frac{c_2 n u^2}{\zeta^2 + u}\right) \tag{67}$$

In particular, the expectation can be bounded as follows.

$$\mathbb{E}\left[Z_n(\zeta)\right] \stackrel{(i)}{\leq} 2\mathbb{E}\left[\sup_{f \in \mathscr{F}: \|f\|_{L_2(\mathcal{D})} \leq \zeta} \frac{1}{n}\left|\sum_i \epsilon_i f(Z_i)\right|\right] \tag{68}$$

$$\stackrel{(ii)}{\leq} 4\text{Rad}_n(\mathscr{F}, \zeta) \leq 4\zeta \delta_n^{\text{dr}} \text{ valid for all } \zeta \geq \delta_n^{\text{dr}} \tag{69}$$

The step (i) follows from symmetrization argument, step (ii) follows from our choice of $\delta_n^{\mathrm{dr}}$.

Thus, we get the desired bound on the event $\mathcal{E}_o$ which completes the proof.

$\square$

## C.3 Instantiating Critical Radius Guarantees for Data-Splitting case

We now provide standard critical radius rates $\delta_n^{\mathrm{ds}}$ for some standard function classes $\mathcal{R}$.

For RKHS classes with eigen vales of kernel $\mathcal{K}$ decaying as $j^{-\frac{1}{\alpha}}$ (eg. Sobolev spaces), the critical radius can be bounded as $O(n^{-\frac{\alpha}{2(\alpha+1)}})$ assuming a bound of unity on the RKHS norm [Wai19].

For VC sub-classes $\mathcal{R}$ (star-shaped at $r_o$) with an absolute bound $S$ on the reward model class, we can bound the critical radius $\delta_n = \sqrt{\frac{d \log n}{n}}$ where $d$ denotes the VC-subgraph dimension [Wai19, CNSS24].

## C.4 Estimation of nuisance parameters

### C.4.1 Plug-in estimates for preference and time

In this case, since the functions $\tanh(x)/x$ is 1-Lipschitz, one can argue that $||\hat{t} - t_o|| \leq ||\hat{\mathfrak{r}} - r_o||$ where $\hat{t} = \frac{\tanh(\hat{\mathfrak{r}})}{\hat{\mathfrak{r}}}$.

One can thus estimate $\mathfrak{r}(\cdot)$ by minimizing the log-loss (3) and since log-loss is strongly convex with parameter $e^{-S}$ [Wai19, Example 14.18] we argue that rate $||\hat{\mathfrak{r}} - r_o||_{L_2(\mathcal{D})} = O\left(e^S \delta_n^2\right)$ where $\delta_n$ is the critical radius of the function class $\mathcal{R}$. Via plugging in i.e. estimating $\hat{t} = \frac{\tanh \mathfrak{r}(\cdot)}{\mathfrak{r}(\cdot)}$, we can argue from Lipschitzness that the same rate holds for $||\hat{t} - t_o||_{L_2(\mathcal{D})}$. However, the estimate $||\hat{\check{t}} - \check{t}_o||$ would suffer from a small bias bounded by $||\check{t}_o - t_o||_{L_\infty}$ which decays with the bound $\check{B}$ as discussed in (47).

### C.4.2 Separate estimation of preference and time

One can get bounds on the rate $||\hat{\check{t}} - \check{t}_o||$ based on the critical radius $\delta_n$ of the class $\mathcal{T}$ post centering at $\check{t}_o$. We can get standard rates for RKHS and VC sub-classes. Estimation of nuisance $\mathfrak{r}(\cdot)$ can be performed using log-loss.

## C.5 Proof of Theorem 5.1

We can prove Theorem 5.1 very similar to the proof of Theorem C.5 by instantiating [FS23, Theorem 1] to our case.

Assumption 1 (Neymann orthogonality) in [FS23, Section 3.2] for the loss function $\mathcal{L}^{\mathrm{ortho}}$ has been already proven in Lemma 3.1. Assumptions 3 (Higher-order smoothness) and 4 (strong-convexity) from [FS23] can be proven very similar to the proof in Lemma C.3 and Lemma C.4.

# D Experimentation Details

Below we provide the experiment details[7].

**Linear Rewards**   We work with a linear reward model defined by $r_\theta(X^1) = \langle \theta, X^1 \rangle$ and $r_\theta(X^2) = \langle \theta, X^2 \rangle$ for each query pair $(X^1, X^2)$, and denote the ground-truth parameter by $\theta_o$. All queries are restricted to lie on the unit-radius shell: we sample each coordinate independently and then rescale the resulting vector so that $\|X\|_2 = 1$. Preferences $Y$ and response times $T$ are generated synthetically according to the EZ diffusion model, producing the dataset $\left\{X_i^1, X_i^2, Y_i, T_i\right\}_{i=1}^n$. Experiments are performed for various values of the norm $B = \|\theta_o\|_2$.

---

[7]The anonymized code can be found at

The ground-truth parameter $\theta_o \in \mathbb{R}^d$ is itself drawn at random, sampling each coordinate independently; every draw therefore yields a new dataset. Our aim is to recover $\theta_o$ using the three losses introduced earlier: the logistic loss $\mathcal{L}^{\mathrm{log}}$, the non-orthogonal loss $\mathcal{L}^{\mathrm{non\text{-}ortho}}$, and the orthogonal loss $\mathcal{L}^{\mathrm{ortho}}$ defined in Equations (3), (4) and (6). For each loss we compute the $\ell_2$ estimation error $\|\hat{\theta} - \theta_o\|_2$ as a function of the true-parameter norm $B = \|\theta_o\|_2$, averaged over 10 datasets generated from different random draws of $\theta_o$. In the orthogonal and non-orthogonal settings the nuisance functions—the reward model $\hat{r}(\cdot)$ and the time model $\hat{t}(\cdot)$—are learned on a held-out split and then plugged into the second-stage optimization. Because $\mathbb{E}[T \mid X]$ is non-linear, we approximate it with a three-layer neural network. For the logistic baseline, the entire dataset is instead used to fit the reward model via standard logistic regression.

**Non-linear rewards—neural networks.** We generate synthetic data from random three-layer neural networks with sigmoid activations in the two hidden layers (widths 64 and 32) and a final linear output layer, fixed input dimension $d = 10$. For each training size $N$, we sample three independent "true" reward networks by drawing each hidden-layer weight matrix and the final-layer vector i.i.d. from $\mathcal{N}(0, 1)$, then for each network generate $N$ training pairs and 3000 test pairs of queries $(X_1, X_2)$ as i.i.d. Gaussian vectors; each reward model is trained four times to assess variability. We evaluate all three losses—logistic, non-orthogonal, and orthogonal—and for the orthogonal loss compare both a simple data-split implementation and a data-reuse implementation. Using this synthetic data, we first learn the nuisance $\mathfrak{r}$ by minimizing the logistic loss with a three-layer network of widths $(10, 32, 16, 1)$, and learn the $t$-nuisance by minimizing squared error on $T$ with a three-layer network of widths $(20, 32, 16, 1)$ taking $(X_1, X_2)$ concatenated as input. Finally, for each repetition we fit the reward model (same architecture as $\mathfrak{r}$) by minimizing each candidate loss. Figure 2 reports the mean squared error of the estimated reward under each loss and the corresponding policy regret after thresholding $\hat{r}$ into a binary decision.

**Text-to-image preference learning.** We evaluate our approach on a real-world text-to-image preference dataset - Pick-a-pick [KPS+23], which contains an approx 500k text-to-image dataset generated from several diffusion models. Furthermore, we use the PickScore model [KPS+23] as an oracle reward function, we simulate binary preferences $Y \in \{+1, -1\}$ and response times $T$ via the EZ-diffusion process conditioned on the PickScore difference between each image-test pair. To learn the reward model we extract 1024-dimensional embeddings from both the text prompt and the generated image using the CLIP model [RKH+21]. On top of these embeddings, we train a 4-layered feed-forward neural network with hidden layers of sizes $1024, 512, 256$, under three training objectives: our proposed orthogonal loss, a non-orthogonal response-time loss, and the standard log-loss on binary preferences. The time nuisance model $t$ uses the same four-layer architecture, and the initial reward nuisance $\mathfrak{r}$ matches the architecture of $r$. For each training size $N$, we draw $N$ random image–text pairs for training and an additional 10000 for testing (from the remaining dataset). For each $N$ we repeat the training process 3 times with different seeds.

### D.1 Additional experiments comparing with [LZR+24]

Our primary focus is supervised reward estimation under passive (i.i.d.) queries, which is often more challenging than adaptive best-arm identification (BAI). To demonstrate improved performance in the adaptive setting as well, we embed our estimator into the BAI pipeline of [LZR+24, Algorithm 1] and compare directly.

**Setup.** We use the real-world food-preference dataset [SK18], which provides pairwise choices and response times (RT) from 42 participants. Following [LZR+24], we construct bandit tasks for each participant and adhere to their sequential-elimination protocol (Algorithm 1 in [LZR+24]).

**Protocol.** We replace their RT estimator with our orthogonal-loss estimator while keeping all other components unchanged. For each participant, we run 70 independent trials at total sample budgets $500, 1000$, and $1500$. As in Fig. 4 of [LZR+24], we summarize the probability of correct selection across participants by reporting the 25th percentile (Q1), median, and 75th percentile (Q3).

**Findings.** Across budgets, our estimator achieves higher probability of correctly identifying the best arm, with the largest gains appearing at smaller budgets; the gap narrows as the budget increases. Full quartile summaries for each budget are reported in Table 3.

**Budget = 500**

| Method | Q1 | Median | Q3 |
|---|---|---|---|
| Preference Only | 0.50 | 0.67 | 0.83 |
| LZR+24 | 0.25 | 0.43 | 0.57 |
| Ours | 0.03 | 0.27 | 0.39 |

**Budget = 1000**

| Method | Q1 | Median | Q3 |
|---|---|---|---|
| Preference Only | 0.37 | 0.56 | 0.64 |
| LZR+24 | 0.03 | 0.10 | 0.37 |
| Ours | 0.00 | 0.10 | 0.26 |

**Budget = 1500**

| Method | Q1 | Median | Q3 |
|---|---|---|---|
| Preference Only | 0.21 | 0.39 | 0.50 |
| LZR+24 | 0.00 | 0.06 | 0.21 |
| Ours | 0.00 | 0.05 | 0.36 |

Table 3: Probability of correctly incorrectly identifying the best arm summarized across participants (Q1/Median/Q3) for the three total sample budgets.

