# OpenReview forum: "Preference Learning with Response Time: Robust Losses and Guarantees"
_NeurIPS.cc/2025/Conference — NeurIPS 2025 poster_

### Official Review · Reviewer_DNMn · 2025-06-30

**Clarity:** 4
**Significance:** 3
**Originality:** 3
**Rating:** 5
**Confidence:** 4

**Summary:**

The authors provide a method to incorporate response time for binary human preference learning, focusing on using basic drift diffusion dynamics to better learn a latent preference function. The EZ drift diffusion model provides expectations of binary responses as a function of stimulus input and reward ($\mathbb{E}[Y|r(X)]$) as well as expectations of response time given input ($\mathbb{E}[T|X]$), allowing for the incorporation of these moments into a preference learning framework. The authors introduce a new neyman-orthogonal loss function to efficiently estimate the latent preference function given response times. They demonstrate better estimates of true underlying reward function using their response-time informed model with their orthogonal loss function using a variety of simulated and semi-simulated datasets. They provide good theoretic properties when reward functions r(X) are linear as well as sampling bounds in general for their estimator. Their theoretical guarantees are impressive and provide an important contribution to this research space.

**Questions:**

My primary question with this work is how it adapts to a GP-based non-parametric logistic settings and in particular the response time GP model mentioned above.

Evaluation on real human response-time data would also increase the impact of this paper. The results of the paper (including the semi-simulated datasets evaluation) implicitly rest on the EZ diffusion time being an accurate model of human response times, but it may not be, especially in certain types of preference learning settings. However, given the large theoretical contribution of their simple assumptions here, is in my opinion not absolutely necessary to evaluate on real response-time data to warrant publication.

One smaller question I have is to what extent the data-splitting and cross-fitting approaches effect the model outcomes. Are the results shown in the figures in the main paper data-splitting or cross-fitting? Some clarity/elaboration here would be helpful.

**Ethical Concerns:**

["NO or VERY MINOR ethics concerns only"]

**Final Justification:**

With the proposed edits I have adjusted my score accordingly. I thank the authors for their thorough response to my concerns.

The authors have included a comparison with the relevant manuscript, so I have further adjusted my score.

**Limitations:**

Some more discussion about the limitations of the method, especially with respect to types of datasets or failure modes of optimization would be helpful. Are there regimes where the overall estimation of r(X) fails? What are the limitations of the EZ diffusion model in capturing real human response times?

**Quality:**

3

**Strengths And Weaknesses:**

The paper is clear and addresses an important set of questions in human preference learning. The analysis is thorough and the main text and especially supplement provides many mathematical properties of their Meta algorithm under data-splitting and cross-fitting approaches. The results on simulated data are compelling and demonstrate the utility of the Neyman-Orthogonal objective, which is an interesting application that is highly applicable in this setting.

However, two of their core contributions, using the DDM to integrate with binary responses and extension to non-parametric reward functions have been previously implemented in a binary preference learning with response time setting (See Shvartsman et al.*) In this paper, the authors very similarly use the properties of the drift diffusion model which has closed-form expressions of moments of the response time distribution as a function of the drift parameter to better estimate a latent preference function. This work was borne out of GP models for human preference learning that did not use response times **, ***. Lack of discussion and comparison with this work as well as the context of this literature severely limits it's impact, as the works are highly related. The primary important distinction is the authors in shvartsman et al only consider non-parametric (GP based) preference functions, but they additionally consider pairwise comparisons with a specific pairwise GP kernel. Comparing their approach the one outlined here is therefore important. Also, the title of this paper "Preference Learning with Response Times" I believe should be changed in light of the fact that this has been done before in Shvartsman et al. The Neyman orthogonal objective and assumption of EZ diffusion process allows for a greater amount of mathematical developman than in Shvartsman et al, and this should be emphasized.

There is no evaluation on real human reaction time data. This makes it difficult to know how reasonable it is to assume the EZ diffusion model holds in a variety of real world preference learning settings.

*Response Time Improves Gaussian Process Models for Perception and Preferences  - Shvartsman, et al. UAI 2024.
** A semi-parametric model for decision making in high-dimensional sensory discrimination tasks. Keeley et. al AAAI 2023
*** Bayesian Active Model Selection with an Application to Automated Audiometry - Gardner et al. NIPS 2015

---

> ### Author Rebuttal · Authors · 2025-07-31
>
> We thank the reviewer for highlighting the important work of Shvartsman et al. While our contributions are distinctly different, we agree that a direct comparison belongs in our work, and we will update our paper with a dedicated comparison with their work. In light of this prior work, we will update the title of the paper to "Robust Losses for Preference Learning with Response Time" in the final version.
>
> While both papers leverage closed‑form DDM moments, our contributions and intended applications are complementary. Specifically:
>
>
> 1. Scalability to high-dimensional inputs.
> Our setting is motivated by large-scale “learning from human feedback” pipelines—e.g., fine‑tuning large language or vision models—where inputs live in very high dimensions and downstream tasks require a pointwise reward estimate. Shvartsman et al. use a GP prior on rewards and develop a variational approximate Bayesian inference approach using moment matching to refine the posterior so as to incorporate response time information. In particular, the paper suffers from the standard issues with GP regression and does not scale to high-dimensional (curse of dimensionality in GP regression [1,2,3]) and is not suitable for large-scale machine learning models.
> We completely bypass the use of variational approximations and moment-matching techniques as our method avoids modeling the full likelihood altogether.   We develop a statistical learning theory approach with a Neyman orthogonal (aka locally robust) loss function that allows for flexible function approximation that can scale large input dimensions and function spaces.
>
> 2. Theoretical guarantees for reward learning.
> We are the first to show that incorporating response times can provably reduce estimation error in reward models in a supervised learning setting. In doing so, we also bridge a gap in prior DDM parameter‑estimation literature and modern learning from human feedback settings by deriving a loss function that admits finite‑sample error bounds for a wide range of function classes.
>
> 3. Function‑class‑agnostic method.
> Our orthogonal loss wraps around any reward learner—neural nets, kernels, or even Bayesian models—allowing practitioners to plug in any approximator and optimize via standard gradient methods.  This agnosticism also paves the way to online learning (e.g., logistic bandits) or implicit reward‑learning (direct policy optimization as described in Appendix A) as future directions.
>
>
> **Regarding adaptability to the response time model in Shvartsman et al**.
>
> Aside from an initial‑state offset $x_o$ of the diffusion process, the DDM in Shvartsman et al. is mathematically equivalent to ours. We assume that the queries are symmetric, that is, swapping the queries $X^1$ and $X^2$ does not affect the model.  In fact, enforcing symmetry—$\text{Pr}(Y=1│X_1=a, X_2 = b) =1–\text{Pr}(Y=1│X_1= b, X_2 =a)$—forces $x_o=0$, yielding exactly the DDM formulation (EZ- diffusion model) in our work. Although our current work does not require it, our orthogonal-loss construction can be extended to accommodate a nonzero initial-state offset.
>
> **Regarding different approaches for nuisance estimation: data-splitting, cross-fitting, data reuse**.
> In the experiments for linear functions, we use data splitting (highlighted in the linear paragraph section in Appendix D) in the main paper. For the case of linear rewards, our preliminary results using data-reuse and cross-fitting show that data-reuse consistently outperforms both cross-fitting and standard data-splitting. In the paper, to demonstrate the benefit of orthogonal loss over preference only estimator for the linear case, we stuck to the more conservative approach of data splitting. We will include a more detailed discussion in the appendix of the experimental section. We ran our neural network experiments with data reuse and data splitting. We consistently observed that data reuse outperformed the data splitting estimator.
>
>
>
> **Regarding methodology scope and limitations**.
> Our method presumes that the drift‐diffusion model (DDM) provides an accurate account of human decision‐making in the target application. When this holds, our orthogonal‐loss framework is highly effective for reward estimation—especially at modern, large scale—but it is important to recognize that DDMs may not capture every nuance of response‐time variability [4,5]. In particular, the EZ diffusion model assumes a constant drift rate and fixed decision boundaries, yet real‐world fluctuations in attention and focus can violate these assumptions [6]. In scenarios where response‐time data depart substantially from the EZ‐DM’s stationary‐drift or boundary‐separation assumptions, our approach may be less appropriate. There exist extensions that relax these constraints [5], and adapting our loss formulation to such generalized diffusion frameworks is an interesting direction for future work.
>
> [1] Giordano, M., Ray, K., & Schmidt-Hieber, J. (2022). On the inability of Gaussian process regression to optimally learn compositional functions. Advances in Neural Information Processing Systems.
> [2] Binois, M., & Wycoff, N. (2022). A survey on high-dimensional Gaussian process modeling with application to Bayesian optimization. ACM Transactions on Evolutionary Learning and Optimization.
> [3] Krauth, K., Bonilla, E. V., Cutajar, K., & Filippone, M. (2016). AutoGP: Exploring the capabilities and limitations of Gaussian process models.
> [4] Ratcliff, R., Smith, P. L., Brown, S. D., & McKoon, G. (2016). Diffusion decision model: Current issues and history. Trends in cognitive sciences.
> [5] Fudenberg, D., Newey, W., Strack, P., & Strzalecki, T. (2020). Testing the drift-diffusion model. Proceedings of the National Academy of Science.
> [6] Myers, C. E., Interian, A., & Moustafa, A. A. (2022). A practical introduction to using the drift diffusion model of decision-making in cognitive psychology, neuroscience, and health sciences. Frontiers in Psychology.

---

### Official Review · Reviewer_iNrS · 2025-07-02

**Clarity:** 3
**Significance:** 2
**Originality:** 3
**Rating:** 4
**Confidence:** 3

**Summary:**

This paper studies an interesting topic where the response time is included for preference learning. It is modeled by the Evidence Accumulation Drift Diffusion, where the response time inversely reflects the preference strength. Based on this model, the authors develop Neyman-orthogonal loss functions and prove that it enjoys oracle convergence rates for linear reward models. The proposed loss function also empirically outperforms log-loss and non-orthogonal loss on different settings: linear reward model and MLP reward model on synthetic dataset; MLP reward model on a text-to-image dataset.

**Questions:**

please see above

**Ethical Concerns:**

["NO or VERY MINOR ethics concerns only"]

**Final Justification:**

The authors' rebuttal answers my first question. I still have concerns on the scalability of the method given the presented experiments in the paper and authors' rebuttal. I hence maintain my weak accept decision.

**Limitations:**

yes

**Quality:**

3

**Strengths And Weaknesses:**

strengths:

- this paper is well-motivated: response time may encode preference strength and we should leverage that during preference learning
- the theoretical analysis on the proposed orthogonal loss and comparison to non-ortho loss is clear and sound
- the empirical experiment design is clean, and the results support the authors' claim

weakness (more like questions):

- since the orthogonal loss requires a preliminary estimator of the true reward function, its estimation error also matters. around line 237, when the preliminary estimator is poorly estimated, you then need an accurate estimate of t to make the product of errors small? i'm wondering how largely this assumption can be true?
- the experiments mostly use simple functions (linear or MLP) and only tracks the loss. in many real world application (e.g., LLM preference learning), we use large transformers for reward modeling. I am wondering if the proposed method can be applied there and how much improvement we can expect over the traditional BT model, especially for the down-stream performance that we really care - the model behavior after preference-tuning?

---

> ### Author Rebuttal · Authors · 2025-07-31
>
> We thank the reviewer for the feedback. We address the comments and questions below:
>
> 1. **Regarding estimation of nuisance function and robustness of preliminary reward error**.
> Note that, since our Neyman‑orthogonal loss is first‑order insensitive to errors in both nuisance functions (the preliminary reward estimate and the response‑time model), one can tolerate significantly slower convergence in the preliminary estimator. For example, an $n^{-1/4}$ rate suffices to achieve $n^{-1/2}$ error rate in reward estimation for various function classes (such as RKHS or parametric models).
> Moreover, as the reviewer points out, the loss is in fact robust to estimation of the preliminary reward function.  In particular, because the preliminary‑reward error only enters multiplied by the error in $t_o$, one may have arbitrarily slow convergence for the preliminary reward estimator so long as the response‑time estimator achieves an $n^{−1/2}$ (or the oracle convergence rate). For example, see the case of linear reward (line 219 (8)).
>
> 2. **Regarding performance in downstream tasks**.
> In our experiments, in addition to MSE loss, we also measure reward‐estimation quality in terms of policy value (see lines 288–289). Although this policy simply selects the option with the higher estimated reward, it serves as a proxy for how a large‑scale policy (e.g., a generative model) would behave after preference tuning. Figures 2 and 3 present these results, showing a clear improvement in policy value.

---

### Official Review · Reviewer_iuYD · 2025-07-02

**Clarity:** 3
**Significance:** 2
**Originality:** 3
**Rating:** 4
**Confidence:** 3

**Summary:**

This paper considers a new modeling approach for preference learning by adding a response time feature into the Bradley-Terry framework. Leveraging EZ diffusion, a Neyman-orthogonal loss function is proposed, and theoretical convergence is derived under both linear reward and general reward scenarios given good nuisance estimation. Experiments on linear, neural-network-based, and synthetic text-to-image reward models show lower MSE/regret with the proposed loss.

**Questions:**

Please refer to weaknesses.

**Ethical Concerns:**

["NO or VERY MINOR ethics concerns only"]

**Final Justification:**

I have looked over the responses to my own questions and identified weaknesses, as well as the other reviews and their responses, which have partially adressed my concerns. I am thus raising my score over the accept line to 4.

**Limitations:**

yes

**Paper Formatting Concerns:**

Formatting concerns: Figure 3 is put on page 10.

**Quality:**

3

**Strengths And Weaknesses:**

*Strengths.*
The theoretical response-time incorporation is well-motivated and supported by the EZ diffusion framework.
The Neyman-orthogonal loss reduces the error during nuisance estimation by zeroing out the first-order impact during parameter estimation and also demonstrates better convergence rates than the logistic loss in theory.
Theoretical results are strong and hold for both linear cases and nonparametric reward classes with either asymptotic or finite-sample guarantees.

*Weaknesses.*
Whether the theoretical framework can be used to describe real-world preferences is not validated, and thus questions the scope of the methodology for application. A plug-in strategy is described in the paper but not validated in realistic setups.
When the theoretical framework (EZ diffusion) fits, in theory, the convergence guarantee for both linear/general reward functions requires accurate nuisance estimation as reflected in Eq. (8) and the (R, $\alpha$) norm in Eq. (9). This questions the degree of practical benefits of the proposal due to the strong reliance of a good preliminary estimator of the true reward function. Furthermore, this makes the comparison between the ortho-estimator and log-loss estimator unfair for both theory and experiments, since sometimes the former uses the latter as an initialization, as discussed in Sections 4 and 6.

---

> ### Author Rebuttal · Authors · 2025-07-31
>
> We thank the reviewer for the feedback. We address the comments and questions below:
>
> 1. **Regarding accurate estimation of Nuisance functions**.
> One of the core advantages of our Neyman‑orthogonal loss is that it is first‑order insensitive to errors in both nuisance functions (the preliminary reward estimate and the response‑time model). In fact, the estimate of the initial reward function can be arbitrarily poor if the estimate of $t_o(X)$ is $\sqrt{n}$ consistent.
> Furthermore, Theorem 5.1, implies that we only need slow rates for the nuisance functions to achieve the oracle rates; nuisance estimators can converge at rates orders of magnitude slower than the oracle rate.  For example, for certain RKHS settings, one still obtains a $\frac{1}{\sqrt{n}}$ convergence rate even when the preliminary reward estimate and the response‑time estimate both converge at a much slower rate of $\frac{1}{n^{1/4}}$.
> Similarly, in equation (8) a $\frac{1}{n^{1/4}}$ rate suffices and can be easily achieved [7] (see the discussion on lines 220-224).
>
> 2. **Regarding fair comparison of log-loss estimator and ortho-estimator**.
> Our goal is *not* to argue that the log‑loss estimator is inferior—when only binary choices are available, log loss is indeed the minimax‐optimal approach. Rather, our focus is on *incorporating response‑time data*, which the log‑loss objective cannot accommodate. We include the log‑loss estimator purely as a baseline to quantify the gain from response‑time augmentation. Initializing our orthogonal estimator with the log‑loss solution speeds up convergence but confers no statistical advantage: all performance improvements stem from the additional information in T.
>
> 3. **Regarding scope of the methodology**.
> Our work targets large‑scale  machine learning —characterized by high dimensional input and large datasets. To emulate real‑world conditions at this scale, we use semi‑synthetic data targeting this regime and work with image and text datasets. Fully real data experiments in this regime would be prohibitively expensive.
> At its core, our work is driven by the practical challenge of integrating drift‑diffusion models into modern, large‑scale machine‑learning pipelines. Whereas classical preference learners (e.g. Bradley–Terry) rely on a tractable likelihood and plug directly into gradient‑based training via maximum‑likelihood estimation, the DDM’s full likelihood is intractable—making it unclear how to exploit response time at scale. Our paper addresses this important gap.
>   Our method presumes that the drift‐diffusion model (EZ-DM) provides an accurate account of human decision‐making in the target application. When this holds, our orthogonal‐loss framework is highly effective for reward estimation. We recognize that DDMs may not capture every nuance of response‐time variability [4,5]. In particular, the EZ diffusion model assumes a constant drift rate and fixed decision boundaries, yet real‐world fluctuations in attention and focus can violate these assumptions [6]. In scenarios where response‐time data depart substantially from the EZ‐DM’s stationary‐drift or boundary‐separation assumptions, our approach may be less appropriate. There exist extensions that relax these constraints [5], and adapting our loss formulation to such generalized diffusion frameworks is an interesting direction for future work.
>
>
> [4] Ratcliff, R., Smith, P. L., Brown, S. D., & McKoon, G. (2016). Diffusion decision model: Current issues and history. Trends in cognitive sciences.
> [5] Fudenberg, D., Newey, W., Strack, P., & Strzalecki, T. (2020). Testing the drift-diffusion model. Proceedings of the National Academy of Science.
> [6] Myers, C. E., Interian, A., & Moustafa, A. A. (2022). A practical introduction to using the drift diffusion model of decision-making in cognitive psychology, neuroscience, and health sciences. Frontiers in Psychology.
> [7] Ji, Ziwei, and Matus Telgarsky. "Risk and parameter convergence of logistic regression." arXiv preprint arXiv:1803.07300 (2018).

---

> > ### Comment · Reviewer_iuYD · 2025-08-05
> > **Thanks**
> >
> > I have read the responses to mine, and the other reviewer's questions and thank the authors for clarifying a number of points. I will raise my score.

---

### Official Review · Reviewer_j9qd · 2025-07-02

**Clarity:** 3
**Significance:** 4
**Originality:** 4
**Rating:** 4
**Confidence:** 3

**Summary:**

This paper introduces a new method for learning human reward models by incorporating user response time into the preference learning process. The central idea is that the time a person takes to choose between two options reveals the strength of their preference, a concept modeled by the Evidence Accumulation Drift Diffusion (EZ) model. While traditional methods rely only on the binary choice (which option was preferred), this work leverages the temporal data of the decision to improve the accuracy and efficiency of reward model learning.

**Questions:**

1. In the orthogonal Loss meta algorithm 1, since they are estimating two more estimators, what is the computational complexity for this algorithm compared to the original loss? Can the authors provide theoretical or experimental evidence?

**Ethical Concerns:**

["NO or VERY MINOR ethics concerns only"]

**Final Justification:**

The authors basically answer my questions, so I will keep the score.

**Limitations:**

See weaknesses.

**Quality:**

3

**Strengths And Weaknesses:**

Strenghts: This work novelly proposes a new preference model by considering response time, and provide rigorous theoretical proof to derive convergence guarantees. Moreover, they proposes an orthogonal loss function and successfully turn from exponentail dependence on the reward range to logarithmic.

Weaknesses:
1. The intuition and the connection between their proposed preference learning with response time and RLHF are not so obvious. Since in RLHF, a standard formulation is bt model. In the author's setting, what is the distribution of the response time and why would the probability of one response is preferred than another grows with the response time. The authors should give RLHF examples to justify the reasonableness of their setting.

2. Besides, their experiments are on bandits, but their theory also lacks a connection to the bandit problem. I understand that the novel regression is their contribution, but since their setting is in RLHF. It would be better if they provide bandit regret bounds in the theoretical part. Also, it is interesting if the response time and be combined with the online decision making process.

---

> ### Author Rebuttal · Authors · 2025-07-31
>
> We thank the reviewer for the feedback. We address the comments and questions below:
>
> 1. **Regarding intuition and connections to the Bradley-Terry model and example application**.
> Note that the EZ diffusion model naturally incorporates the Bradley Terry model for preferences (see equation 1 in the paper). Crucially, the EZ diffusion model allows to exploit the response time information in learning the reward model. In practical settings, the response time can be cheaply recorded along with preferences and are indicative of preference strength (see the discussion on lines 43-54 for a detailed discussion). A key application is fine-tuning diffusion models with human preference data [c], which motivates our text- and image-based experiments in Figure 3.
>
> 2. **Regarding the distribution of response time**.
> One of the core contributions of our paper is handling the intractable likelihood of the response time. The likelihood of the response time is reduced to an infinite sum [a], and prior works approximate with either a parametrized heavy-tailed distribution [b] or series truncation [a]. In contrast, our Neyman-orthogonal loss entirely obviates likelihood computation, relying only on the closed-form conditional expectation.
>
> 3. **Regarding bandit regret bounds**.
> Our framework operates in a standard supervised-learning setting rather than an adaptive bandit environment. In our experiments, we evaluate our method by the MSE of reward estimation, and—as a proxy for downstream tasks—compare the value of the naïve policy that always picks the action with the highest estimated reward.
> This crucially differentiates us from prior work in [LZR+24] and addresses a major drawback of the prior approach. We tackle a more general—and inherently harder—supervised learning problem without adaptive query selection. The queries are assumed to be coming from an unknown distribution. While one could extend our approach to active query selection (e.g., optimal experiment design) and recover a bandit‐style analysis in the linear case—yielding guarantees similar to [LZR+24]—this is beyond the paper’s current scope. In practice, even for the case of the best arm selection problem, we observe that our method generally has a superior performance to [LZR+24]. For a more detailed discussion, please see our response to reviewer rfdo.
>
> 4. **Regarding computational complexity**.
> Our meta‐algorithm supports any function class and gradient‐based optimizer. Its runtime scales with the chosen optimizer, the model class, and the dataset size, but remains asymptotically identical to the preference‐only estimator. In particular, when instantiated with a linear reward model, the optimization is convex and thus solvable in polynomial time. Empirically, for neural network experiments, nuisance estimation and orthogonal-loss minimization require similar runtimes.
>
> [a] Navarro, D. J., & Fuss, I. G. (2009). Fast and accurate calculations for first-passage times in Wiener diffusion models. Journal of mathematical psychology.
> [b] Shvartsman, M., Letham, B., & Keeley, S. (2023). Response Time Improves Choice Prediction and Function Estimation for Gaussian Process Models of Perception and Preferences.
> [c] Wu, X., Sun, K., Zhu, F., Zhao, R., & Li, H. (2023). Human preference score: Better aligning text-to-image models with human preference. In Proceedings of the IEEE/CVF International Conference on Computer Vision.

---

> > ### Author Response · Authors · 2025-08-06
> > **Rebuttal follow-up**
> >
> > We thank the reviewer again for their time and review. We kindly request that they respond to our rebuttal and let us know if there are any remaining questions or concerns.

---

### Official Review · Reviewer_rfdo · 2025-07-03

**Clarity:** 3
**Significance:** 3
**Originality:** 2
**Rating:** 5
**Confidence:** 2

**Summary:**

This paper considers using the EZ-diffusion model — a model of the binary choice process that models response times — for learning from preferences. The advantage of this model is that the observed response time for a given choice reveals additional information about the reward difference between the two options. The downside is that because both the expected response time and the reward difference must now be estimated, there is a risk that a biased estimate of the expected response time will also bias the estimated reward difference. This paper addresses this by introducing a new Neyman-orthogonal loss function for this setting, thus largely avoiding response time mis-estimates biasing the reward estimate.

The paper then proves a number of asymptotic and finite-sample bounds on the reward function inference error. Empirical experiments (with synthetic choice generated from the EZ-diffusion model) show that accounting for response times yields better reward inference than a logistic model (which does not model response times), and that the proposed orthogonal loss yields a decent further improvement in inference quality.

**Questions:**

On line 96: does $Y = +1$ signify a choice for $X^1$ or $X^2$? This is not actually stated.

**Ethical Concerns:**

["NO or VERY MINOR ethics concerns only"]

**Final Justification:**

As stated in my original review, the paper is technically solid and provides valuable theoretical insights. My initial concern was mostly that the method had not been tested on human data, but this has now been addressed by the authors.

**Limitations:**

No specific section on the limitations of the proposed method is present. There are obvious limitations of the work, for example the asymptotic guarantees being limited to linear reward functions, which should have been listed. There are also limitations inherent in using the EZ-diffusion model for choice modeling (such as needing to record response times), which are of course shared with prior work, but should have been listed nonetheless.

**Quality:**

3

**Strengths And Weaknesses:**

The proposed orthogonal loss yields (provable!) improvements in reward inferences from human preferences when response times are available. The loss itself requires little additional work compared to baselines and is therefore an important and valuable contribution to the field.

The theoretical results help to show (more so than empirical evaluation) the benefits of the proposed loss. The asymptotic analysis only applies to linear reward functions but this is nevertheless an important class of reward functions which is widely used in practice. As someone who works with preference learning and human choice modeling in a more practical capacity, the finite sample analysis in section 5 was a bit beyond my mathematical background, but I do appreciate the headline conclusions here.

Though I have no doubt that the theoretical work done in this paper yield a sufficiently large contribution for acceptance, I do find the empirical experiments to be lacking. The motivation for this paper is to improve learning from human preferences, and data sets of human preference with response times exist (see e.g. the data used in [LZR+24]), so why was that data not used in the experiments here?

Finally, I do think the authors should address the lack of a conclusion (highly unusual) or a discussion on limitations (absolutely necessary).

---

> ### Author Rebuttal · Authors · 2025-07-30
>
> We thank the reviewer for the feedback. We address the comments and questions below:
>
> 1. **Additional experiments based on [LZR+24]**.
> We tackle a more general—and inherently harder—supervised learning problem without adaptive query selection, yet even in the simpler adaptive best-arm identification setting, our method outperforms the estimator from [LZR⁺24].  We evaluate our orthogonal-loss estimator on bandit tasks derived from the real-world food-preference dataset [1], which contains choice and response-time data from 42 participants. Following their sequential-elimination protocol (Algorithm 1 in [LZR⁺24]), we substitute our orthogonal estimator for their response time estimator, run 70 independent trials per participant, and then summarize the probability of correctly identifying the best arm across participants (Figure 4 in [LZR+24])  via the 25th percentile (Q1), median, and 75th percentile (Q3) for each total sample budget. Results at budgets 500, 1000, and 1500 are shown below. We observe that our estimator clearly outperforms their proposed estimator in [LZR+24] for smaller budget values and the improvement reduces as the budget increases.
>   ---
> *Budget = 500*
>
> | Method          | Q1   | Median | Q3   |
> |:----------------|:-----|:-------|:-----|
> | Preference Only | 0.50 |  0.67  | 0.83 |
> | LZR+24          | 0.25 |  0.43  | 0.57 |
> | Ours            | 0.03 |  0.27  | 0.39 |
>
>
>
> ---
>
> *Budget = 1000*
>
> | Method          | Q1   | Median | Q3   |
> |:----------------|:-----|:-------|:-----|
> | Preference Only | 0.37 |  0.56  | 0.64 |
> | LZR+24          | 0.03 |  0.10  | 0.37 |
> | Ours            | 0.00 |  0.10  | 0.26 |
>
> ---
>
> *Budget = 1500*
>
> | Method          | Q1   | Median | Q3   |
> |:----------------|:-----|:-------|:-----|
> | Preference Only | 0.21 |  0.39  | 0.50 |
> | LZR+24          | 0.00 |  0.06  | 0.21 |
> | Ours            | 0.00 |  0.05  | 0.36 |
> ---
>
>
> 2. **Limitations of our work**.
> Our method presumes that the drift‐diffusion model (EZ-DM) provides an accurate account of human decision‐making in the target application. When this holds, our orthogonal‐loss framework is highly effective for reward estimation. We recognize that DDMs may not capture every nuance of response‐time variability [4,5]. In particular, the EZ diffusion model assumes a constant drift rate and fixed decision boundaries, yet real‐world fluctuations in attention and focus can violate these assumptions [6]. In scenarios where response‐time data depart substantially from the EZ‐DM’s stationary‐drift or boundary‐separation assumptions, our approach may be less appropriate. There exist extensions that relax these constraints [5], and adapting our loss formulation to such generalized diffusion frameworks is an interesting direction for future work.
>
> 3. **Regarding asymptotic guarantees**.
> Our non-asymptotic bounds in Section 5 already imply finite-sample convergence rates for linear reward models. We further derive asymptotic results in the linear setting to emphasize the exponential pointwise improvement over the preference-only estimator. We will clarify this distinction in the updated version.
>
> 4. **"Does $Y=1$ signify a choice for $X^1$ or $X^2$?"**
> Thank you for highlighting this. We will revise the text to state explicitly that $Y=1$ indicates that $X^1$ was chosen.
>
> [1] S. M. Smith and I. Krajbich. Attention and choice across domains. Journal of Experimental Psychology.
> [4] Ratcliff, R., Smith, P. L., Brown, S. D., & McKoon, G. (2016). Diffusion decision model: Current issues and history. Trends in cognitive sciences.
> [5] Fudenberg, D., Newey, W., Strack, P., & Strzalecki, T. (2020). Testing the drift-diffusion model. Proceedings of the National Academy of Science.
> [6] Myers, C. E., Interian, A., & Moustafa, A. A. (2022). A practical introduction to using the drift diffusion model of decision-making in cognitive psychology, neuroscience, and health sciences. Frontiers in Psychology.

---

> > ### Comment · Reviewer_rfdo · 2025-08-04
> >
> > Thank you for the rebuttal.
> >
> > I am slightly confused about the new results. In your explanation here you say that the numbers in the table above represent the probability of selecting the best arm. However, referencing these numbers to Figure 4 in [LZR⁺24] it seems that the numbers actually represent the probability of error; this would make much more sense and aligns with the conclusion you draw here. Is this the case?
> >
> > Assuming these new results are indeed error probabilities (please confirm!), I think with the addition of these results and the discussion of the limitations this is now an excellent paper. I have raised my score accordingly.

---

> ### Author Response · Authors · 2025-08-04
> **Clarification**
>
> We thank the reviewer for increasing the score and judging the paper favorably.
>
> Yes, the numbers in the table represent the probability of error in identifying the best arm, as in [LZR+24]. Our earlier reference to it as identifying the probability of the best arm was indeed a typo.

---

### Decision · Program_Chairs · 2025-09-17

**Decision:**

Accept (poster)

**Comment:**

The reviewers agreed that the paper studies an interesting problem of modeling the response time along with binary preference within frameworks of learning from human preferences. The paper introduces a novel methodology to incorporate response time, and demonstrates its effectiveness in eliciting a more effective reward model through theoretical analysis and experiments. However, the reviewers also raised several concerns and questions in their initial reviews, particularly regarding the limited experimental evaluation. We want to thank the authors for their responses and active engagement during the discussion phase. The reviewers appreciated the responses, which helped in answering their key questions. The reviewers have an overall positive assessment of the paper, and there is a consensus for acceptance. The reviewers have provided detailed feedback, and we strongly encourage the authors to incorporate this feedback when preparing the final version of the paper.